# Non-Rigid Shape Registration via Deep Functional Maps Prior

**Puhua Jiang**[1,2#]    **Mingze Sun**[1#]    **Ruqi Huang**[1*]
1. Tsinghua Shenzhen International Graduate School, China    2. Peng Cheng Lab, China

## Abstract

In this paper, we propose a learning-based framework for non-rigid shape registration *without correspondence supervision*. Traditional shape registration techniques typically rely on correspondences induced by extrinsic proximity, therefore can fail in the presence of large intrinsic deformations. Spectral mapping methods overcome this challenge by embedding shapes into, geometric or learned, high-dimensional spaces, where shapes are easier to align. However, due to the dependency on abstract, non-linear embedding schemes, the latter can be vulnerable with respect to perturbed or alien input. In light of this, our framework takes the best of both worlds. Namely, we deform source mesh towards the target point cloud, guided by correspondences induced by high-dimensional embeddings learned from deep functional maps (DFM). In particular, the correspondences are dynamically updated according to the intermediate registrations and filtered by consistency prior, which prominently robustify the overall pipeline. Moreover, in order to alleviate the requirement of extrinsically aligned input, we train an orientation regressor on a set of aligned synthetic shapes independent of the training shapes for DFM. Empirical results show that, with as few as dozens of training shapes of limited variability, our pipeline achieves state-of-the-art results on several benchmarks of non-rigid point cloud matching, but also delivers high-quality correspondences between unseen challenging shape pairs that undergo both significant extrinsic and intrinsic deformations, in which case neither traditional registration methods nor intrinsic methods work. The code is available at `https://github.com/rqhuang88/DFR`.

## 1 Introduction

Estimating correspondences between deformable shapes is a fundamental task in computer vision and graphics, with a significant impact on an array of applications including robotic vision [50], animation [40], 3D reconstruction [58], to name a few. In this paper, we tackle the challenging task of estimating correspondences between *unstructured* point clouds sampled from surfaces undergoing *significant* non-rigid deformations.

In fact, this task has attracted increasing interest [59; 26; 35; 24; 8] in shape matching community. These methods all follow a data-driven approach to learn embedding schemes, which project point clouds into certain high-dimensional spaces. By lifting to high-dimensional spaces, the non-rigid deformations are better characterized than in the ambient space, $\mathbb{R}^3$. The optimal transformations and therefore the correspondences are then estimated in the embedded spaces. Despite the great progresses, such approaches suffer from the following limitations: 1) Their performance on unseen shapes is largely unwarranted; 2) The learned high-dimensional embeddings lack intuitive geometric meaning, and the induced correspondences are difficult to evaluate and analyze without ground-truth labels.

---

[*][#] indicates equal contribution. Corresponding author: ruqihuang@sz.tsinghua.edu.cn

37th Conference on Neural Information Processing Systems (NeurIPS 2023).

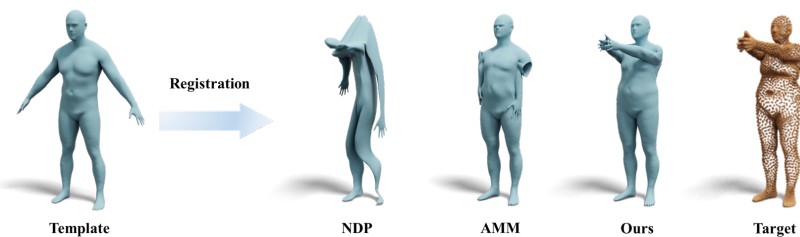

Registration

Template        NDP      AMM      Ours      Target

Figure 1: Shape registration methods like NDP [31] and AMM [55] estimate intermediate correspondences via extrinsic proximity, therefore suffering from large intrinsic deformations. In contrast, by incorporating a DFM-based point feature extractor, our method successfully deforms a FAUST template (the left-most mesh) to another individual of a different pose (the right-most point cloud).

On the other hand, the above limitations are less of a concern from the perspective of shape registration, either axiomatic [1; 55] or learning-based [31]. Unlike the former, the latter deform a source shape to non-rigidly align with a target explicitly in the ambient space $\mathbb{R}^3$. In consequence, the mapping quality can be directly related to the alignment quality. Given a deformed source shape and a target, one can evaluate both qualitatively (by visual comparison) and quantitatively (by computing Chamfer distance), and, more importantly, further optimize the registration outcome via the correspondence-label-free quantitative metric. However, shape registration itself does not serve as a direct rescue to our problem of interest, as the previous approaches typically rely on the premise that the undergoing non-rigid deformation can be approximated by a set of local, small-to-moderate, rigid deformations, which severely hinders their performance in the presence of large deformations (see Fig. 1) and heterogeneity.

Motivated by the above observations, we propose a framework to take the best of classic shape registration and learning-based embedding techniques. In a nutshell, we leverage the estimated correspondence from the latter to guide shape deformation via the former iteratively. Our key insight is to enforce similarity between the deformed source and target in *both ambient space and learned high-dimensional space*. Intuitively, in the presence of large deformation, the learned correspondence is more reliable than that obtained by proximity in the ambient space. On the other hand, properly deforming source mesh to target point cloud increases spatial similarity in the ambient space, leading to higher similarity in the embedded space and therefore more accurate correspondences. In the end, we can compute correspondences between raw point clouds with a shared source mesh as a hub. As will be shown in Sec. 4, our method allows for choosing source shape during inference, which can be independent of the training data.

While being conceptually straightforward, we strive to improve the performance, robustness and practical utility of our pipeline by introducing several tailored designs. First of all, the key component of our pipeline is an embedding scheme for accurately and efficiently estimating correspondences between the deformed source shape and the target *point cloud*. In particular, to take advantage of Deep Functional Maps (DFM) [29; 7; 49], the current state-of-the-art approach on matching *triangular meshes*, we pre-train an unsupervised DFM [49] on meshes as a teacher net and then learn a point-based feature extractor (*i.e.*, an embedding scheme for points) as a student net on the corresponding vertex sets with a natural self-supervision. Unlike the approach taken in [8], which heavily relies on DiffusionNet [47] to extract intricate structural details (e.g., Laplace-Beltrami operators) from both mesh and point cloud inputs, the teacher-student paradigm allows us to utilize a more streamlined backbone – DGCNN [53]. Secondly, in contrast to prior works that either implicitly [31; 55] or explicitly [46; 8; 24] require rigidly aligned shapes for input/initialization, or demand dense correspondence labels [35]. We train an orientation regressor on a large set of synthetic, rigidly aligned shapes to *automatically* pre-process input point clouds in arbitrary orientations. Last but not least, we propose a dynamic correspondence updating scheme, bijectivity-based correspondence filtering, two-stage registration in our pipeline. We defer the technical details to Sec. 3.

Overall, our framework enjoys the following properties: 1) Due to the hybrid nature of our framework, it can handle point clouds undergoing significant deformation and/or heterogeneity; 2) As our feature extractor is self-supervised by some deep functional maps network, our framework is free of correspondence annotation throughout; 3) Our core operation is performed in the ambient space, enabling more efficient, faithful, and straightforward analysis of registration/mapping results than the

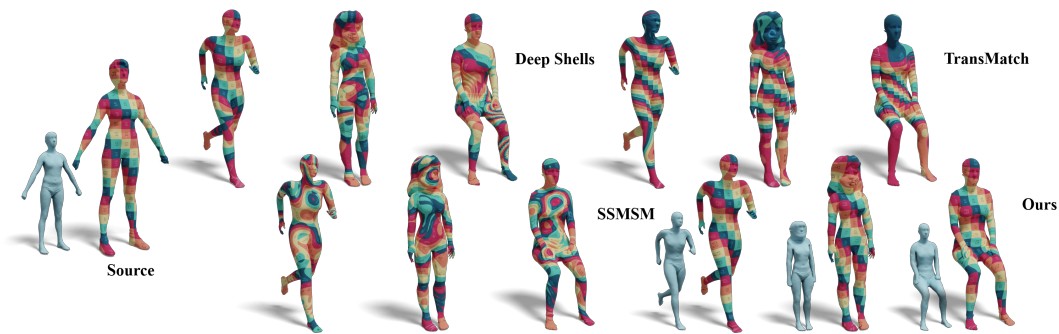

Figure 2: We estimate correspondences between heterogeneous shapes from SHREC'07 with four learning-based methods, all trained on the SCAPE_r dataset. Our method outperforms the competing methods by a large margin. Remarkably, our method manages to deform a SCAPE template shape to heterogeneous shapes, as indicated by the blue shapes.

purely learning-based approaches; 4) Based on a data-driven orientation regressor, we achieve an automatic pipeline for estimating correspondences between deformable point clouds.

As shown in Fig. 2, trained on the SCAPE [2] dataset, our framework generalizes well and outperforms the competing methods by a large margin in the challenging SHREC07-H dataset, which presents prominent variability in both extrinsic orientation and intrinsic geometry. We conduct a set of experiments to verify the effectiveness of our pipeline. We highlight that it achieves state-of-the-art results in matching non-rigid point clouds in both near-isometric and heterogeneous shape collections. More remarkably, it generalizes well despite the distinctiveness between the training set and test set. Moreover, our method, being point-based, is more robust with respect to topological perturbations within input shapes than the mesh-based baselines.

## 2 Related Works

**Non-rigid Shape Registration:** Unlike the rigid counterpart, non-rigidly aligning shapes is more challenging because of the perplexity of deformation models. In general, axiomatic approaches [1] assume the deformation of interest can be approximated by local, small-to-moderate, rigid deformations, therefore suffer from large intrinsic deformations. In fact, recent advances such as [60; 28; 55] mainly focus on improving the efficiency and robustness regarding noises and outliers. On the other hand, there exists as well a trend of incorporating deep learning techniques [6; 5; 21]. Among them, NDP [31] is similar to our method in the sense that the non-learned version follows an optimization-based approach based on some neural encoding. However, its shape registration is purely guided by proximity in the ambient space, hindering its performance in our problem of interest. Perhaps the most relevant approach with our method along this line is TransMatch [51], which follows a supervised learning scheme and learns a transformer to predict directly 3D flows between input point clouds. As demonstrated in Fig. 2 and further in Sec. 4, as a supervised method, TransMatch generalizes poorly to unseen shapes.

**Non-rigid Point Cloud Matching:** Unlike the above, several recent approaches [26; 30; 59] directly establish correspondence between a pair of point clouds. In general, these methods embed point clouds into a canonical feature space, and then estimate correspondences via the respective proximity. Leapard [30] learns such feature extractor under the supervision of ground-truth correspondences between partial point clouds with potentially low overlaps. On the other hand, DPC [26] and CorrnetNet3D [59] leverage point cloud reconstruction as proxy task to learn embeddings without correspondence label. Since intrinsic information is not explicitly formulated in these methods, they can suffer from significant intrinsic deformations and often generalize poorly to unseen shapes.

**Deep Functional Maps:** Stemming from the seminal work of functional maps [39] and a series of follow-ups [38; 22; 44; 37; 23], spectral methods have achieved significant progress in axiomatic non-rigid shape matching problem, and consequentially laid solid foundation for the more recent advances on Deep Functional Maps (DFM), which is pioneered by Litany et al. [32]. Unlike axiomatic functional maps frameworks that aim at optimizing for correspondences represented in spectral basis (so-called functional maps) *with hand-crafted features as prior*. DFM takes an inverse

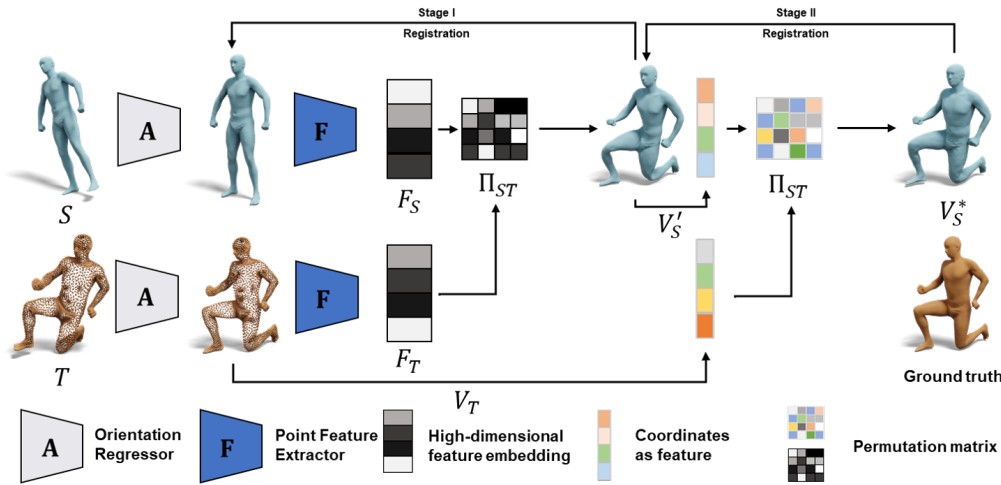

Figure 3: The schematic illustration of our pipeline. **A** is a pre-trained orientation regressor for aligning input shapes. Then a pre-trained feature extractor **F** embeds them into a high-dimensional canonical space. During the iterative optimization procedure of registration, correspondences are dynamically updated according to learned features (Stage-I) and coordinates (Stage-II) of the intermediate shapes. See more details in the text.

viewpoint and aims to search for the optimal features, such that the induced functional maps satisfy certain structural priors as well as possible. Recent findings along this direction empirically suggest that the structural priors suffice to train an effective feature extractor, without any correspondence supervision [12; 29; 7; 3]. However, because of the heavy dependence of eigenbasis of Laplace-Beltrami operators, DFM is primarily designed for shapes represented by triangle meshes and can suffer notable performance drop when applied to point clouds without adaptation [8].

In fact, inspired by the success of DFM, several approaches [24; 8] have been proposed to leverage intrinsic geometry information carried by meshes in the training of feature extractors tailored for non-structural point clouds. Among these, NIE [24] and SSMSM [8] are closely related to our approach. As all three works attempt to train an effective, intrinsic-geometry-aware, per-point feature extractor by leveraging meshes during training. However, unlike the former two, our method incorporates optimization-based shape registration techniques, which effectively alleviates the limitation of purely relying on an abstract high-dimensional embedding scheme (*i.e.*, the learned feature extractor).

**Combination of Extrinsic and Intrinsic Methods:** Unlike the above, our method operates in both extrinsic and intrinsic spaces. From this point of view, the most relevant work is Smooth Shells [14], which proposes to perform shape registration under a mixed representation of shapes, including eigenbasis, coordinates, and normal. Though achieving considerable high-quality correspondence on standard benchmarks, it heavily relies on the availability of mesh representation, but also is sensitive to the original extrinsic alignment of input shapes. These limitations are inherited by Deep Shells [16] – the learning version of the former. Similarly, NeuralMorph [15] jointly performs interpolation and correspondence estimation between a pair of shapes, which inject the intrinsic information by imposing geodesic preservation loss. As a result, it is non-trivial to lift the requirement of mesh input.

## 3 Methodology

Given a pair of shapes $\mathcal{S}, \mathcal{T}$, our target is to deform $\mathcal{S}$ to non-rigidly align with $\mathcal{T}$. We assume that $\mathcal{S}$ is represented as a triangle mesh so that we can effectively regularize the deformed shape by preserving local intrinsic geometry. On the other hand, we require *no* structural information on $\mathcal{T}$ and generally assume it to be a *point cloud*. Our pipeline can also be extended to compute correspondences between two raw point clouds $\mathcal{T}_1, \mathcal{T}_2$. To this end, we fix a template mesh $\mathcal{S}$, perform respective shape registration between $\mathcal{S}$ and the target point clouds, and finally compute the map by composition $T_{12} = T_{s2} \circ T_{1s}$.

Our pipeline is shown in Fig. 3, which consists of three main components: 1) An orientation regressor, **A**, for *extrinsically* aligning input shapes, either mesh or point cloud; 2) A point feature extractor, **F**,

trained under deep functional maps scheme; 3) A registration module that iteratively optimizes for deformations non-rigidly aligning $\mathcal{S}$ with $\mathcal{T}$. In particular, it takes the rigidly aligned shapes from 1) as input, and leverages 2) to update correspondences during optimization.

Though our pipeline leverages a pre-trained feature extractor as registration prior, we highlight that neither the source nor the target is necessarily within or close to the respective training set. We provide throughout experimental evidence in Sec. 4 showing the generalizability of our pipeline.

## 3.1 Orientation Regressor

Our first objective is to align input point clouds with arbitrary orientations into a canonical frame. We take a data-driven approach by training an orientation regressor [9] on 5000 synthetic SURREAL shapes from [52], which are implicitly aligned to a canonical frame by the corresponding generative codes. We refer readers to Supplementary Materials for more details.

## 3.2 Point Feature Extractor

In order to efficiently and accurately estimate correspondences between deformed source shape and the target point cloud during registration, our next goal is to train a feature extractor for point clouds, which is intrinsic-geometry aware. The authors of [8] propose a multi-modal feature extractor based on DiffusionNet [47], which can process point clouds with an extra step of *graph* Laplacian construction [48]. Though it has demonstrated satisfying performance accommodating point cloud representation in [8], the explicit graph Laplacian construction is computationally heavy. Hence, we adopt the modified DGCNN proposed in [24] as our backbone, which is lightweight and robust regarding the sampling density and sizes of point clouds.

In particular, our training follows a teacher-student paradigm. Namely, we first train a deep functional maps (DFM) network, $\mathbf{G}$, on a collection of meshes. Then we train a DFM network $\mathbf{F}$ on the corresponding vertex sets, with an extra self-supervision according to the inclusion map between meshes and their vertex sets. In other words, the feature produced by $\mathbf{F}$ is *point-wisely* aligned with that produced by $\mathbf{G}$.

**Training DFM on Meshes:** Our training scheme in principle follows that of [49], which delivers both accurate map estimation and excellent generalization performance.

1) We take a pair of shapes $\mathcal{S}_1, \mathcal{S}_2$, and compute the leading $k$ eigenfunctions of the Laplace-Beltrami operator on each shape. The eigenfunctions are stored as matrices $\Phi_i \in \mathbb{R}^{n_i \times k}, i = 1, 2$.

2) We compute $G_1 = \mathbf{G}(\mathcal{S}_1), G_2 = \mathbf{G}(\mathcal{S}_2)$, projected into the spaces spanned by $\Phi_1, \Phi_2$ respectively, and leverage the differential FMReg module proposed in [13] to estimate the functional maps $C_{12}, C_{21}$ between the two shapes, then by regularizing the structure of $C_{12}, C_{21}$, say:

$$E_{\text{bij}}(\mathbf{G}) = \|C_{12}C_{21} - I\|_F^2, E_{\text{ortho}}(\mathbf{G}) = \|C_{12}C_{12}^T - I\| + \|C_{21}C_{21}^T - I\|. \tag{1}$$

3) To enhance $\mathbf{G}$, we estimate the correspondences between $\mathcal{S}_1, \mathcal{S}_2$ via the proximity among the rows of $G_1, G_2$ respectively. Namely, we compute

$$\Pi_{12}(i, j) = \frac{\exp(-\alpha \delta_{ij})}{\sum_{j'} \exp(-\alpha \delta_{ij'})}, \forall i \in [|\mathcal{S}_1|], j \in [|\mathcal{S}_2|], \tag{2}$$

where $\delta_{ij} = \|G_1(i, :) - G_2(j, :)\|$ and $\alpha$ is the temperature parameter, which is increased during training [49]. Then $\Pi_{12}$ is the soft map between the two shapes. Ideally, it should be consistent with the functional maps estimated in Eqn.(1), giving rise to the following loss:

$$E_{\text{align}}(\mathbf{G}) = \|C_{12} - \Phi_2^\dagger \Pi_{21} \Phi_1\| + \|C_{21} - \Phi_1^\dagger \Pi_{12} \Phi_2\|, \tag{3}$$

where $\dagger$ denotes to the pseudo-inverse. To sum up, $\mathbf{G}^* = \arg\min E_{\text{DFM}}(\mathbf{G})$, which is defined as:

$$E_{\text{DFM}}(\mathbf{G}) = \lambda_{\text{bij}} E_{\text{bij}}(\mathbf{G}) + \lambda_{\text{orth}} E_{\text{orth}}(\mathbf{G}) + \lambda_{\text{align}} E_{\text{align}}(\mathbf{G}). \tag{4}$$

**Training DFM on Point Clouds:** Now we are ready for training $\mathbf{F}$ with a modified DGCNN [24] as backbone. We let $F_1 = \mathbf{F}(\mathcal{S}_1), F_2 = \mathbf{F}(\mathcal{S}_2)$. On top of $E_{\text{DFM}}$ in Eqn. (4), we further enforce the

extracted feature from $\mathbf{F}$ to be aligned with that from $\mathbf{G}^*$ via PointInfoNCE loss [54]:

$$E_{\text{NCE}}(\mathbf{F}, \mathbf{G}^*) = -\sum_{i=1}^{n_1} \log \frac{\exp(\langle F_1(i,:), G_1^*(i,:)\rangle/\gamma)}{\sum\limits_{j=1}^{n_1} \exp(\langle F_1(i,:), G_1^*(j,:)\rangle/\gamma)} - \sum_{i=1}^{n_2} \log \frac{\exp(\langle F_2(i,:), G_2^*(i,:)\rangle/\gamma)}{\sum\limits_{j=1}^{n_2} \exp(\langle F_2(i,:), G_2^*(j,:)\rangle/\gamma)},$$

(5)

where $\langle . \rangle$ refers to the dot products, $\gamma$ is the temperature parameter, and $n_i$ is the number of points of $\mathcal{S}_i$. So the final training loss is defined as:

$$E(\mathbf{F}) = E_{\text{DFM}}(\mathbf{F}) + \lambda_{\text{NCE}} E_{\text{NCE}}(\mathbf{F}, \mathbf{G}^*),$$

(6)

which is optimized on all pairs of shapes within the training set.

## 3.3 Shape Registration

Now we are ready to formulate our registration module. We first describe the deformation graph construction in our pipeline. Then we formulate the involved energy functions and finally describe the iterative optimization algorithm. During registration, we denote the deformed source model by $\mathcal{S}^k = \{\mathcal{V}^k, \mathcal{E}\}, \mathcal{V}^k = \left\{ v_i^k \mid i = 1, \dots, N \right\}$, where $k$ indicates the iteration index and $v_i^k$ is the position of the $i-$th vertex at iteration $k$. The target point cloud is denoted by $\mathcal{T} = \{u_j \mid j = 1, \dots, M\}$.

**Deformation Graph:** Following[20], we reduce the vertex number on $\mathcal{S}$ to $H = [N/2]$ with QSlim algorithm [17]. Then an embedded deformation graph $\mathcal{DG}$ is parameterized with axis angles $\Theta \in \mathbb{R}^{H \times 3}$ and translations $\Delta \in \mathbb{R}^{H \times 3}$: $\mathbf{X} = \{\Theta, \Delta\}$. The vertex displacements are then computed from the corresponding deformation nodes. For a given $\mathbf{X}^k$, we can compute displaced vertices via:

$$\mathcal{V}^{k+1} = \mathcal{DG}(\mathbf{X}^k, \mathcal{V}^k).$$

(7)

**Dynamic correspondence update:** Thanks to our point-based feature extractor $\mathbf{F}$, we can freely update correspondences between deforming source mesh and target point clouds. In practice, for the sake of efficiency, we update every 100 iterations in optimization (see Alg. 1).

**Correspondence Filtering via Bijectivity:** One key step in shape registration is to update correspondences between the deforming source and the target. It is then critical to reject erroneous corresponding points to prevent error accumulation over iterations. In particular, we propose a filter based on the well-known bijectivity prior. Given $\mathcal{S}^k$ and $\mathcal{T}$, we first compute maps represented as permutation matrices $\Pi_{\mathcal{ST}}, \Pi_{\mathcal{TS}}$, either based on the learned feature or the coordinates. For $v_i^k \in \mathcal{S}^k$, we compute the geodesic on $\mathcal{S}^k$ between $v_i^k$ and its image under permutation $\Pi_{\mathcal{ST}} \Pi_{\mathcal{TS}}$, then reject all $v_i^k$ if the distance is above some threshold. For the sake of efficiency, we pre-compute the geodesic matrix of $\mathcal{S}$ and approximate that of $\mathcal{S}^k$ with it. Finally, we denote the filtered set of correspondences by $\mathcal{C}^k = \{(v_{i_1}^k, u_{j_1}), (v_{i_2}^k, u_{j_2}) \cdots \}$.

In the following, we introduce the energy terms regarding our registration pipeline.

**Correspondence Term** measures the distance of filtered correspondences between $\mathcal{S}^k$ and $\mathcal{T}$, given as:

$$E_{\text{corr}} = \frac{1}{|\mathcal{C}^k|} \sum_{(v_i^k, u_j) \in \mathcal{C}^k} \left\| v_i^k - u_j \right\|_2^2.$$

(8)

**Chamfer Distance Term** has been widely used [31; 30] to measure the extrinsic distance between $\mathcal{S}^k$ and $\mathcal{T}$:

$$E_{\text{cd}} = \frac{1}{N} \sum_{i \in [N]} \min_{j \in [M]} \left\| v_i^k - u_j \right\|_2^2 + \frac{1}{M} \sum_{j \in [M]} \min_{i \in [N]} \left\| v_i^k - u_j \right\|_2^2$$

(9)

**As-rigid-as-possible Term** reflects the deviation of estimated local surface deformations from rigid transformations. We follow [20; 27] and define it as:

$$E_{\text{arap}} = \sum_{h \in [H]} \sum_{l \in \psi(h)} (\left\| d_{h,l}(\mathbf{X}) \right\|_2^2 + \alpha \left\| (R(\Theta_h) - R(\Theta_l) \right\|_2^2)$$

(10)

**Algorithm 1:** Shape registration pipeline.

---

**Input:** Source mesh $\mathcal{S} = \{\mathcal{V}, \mathcal{E}\}$ and target point cloud $\mathcal{T}$; Trained point feature extractor $\mathbf{F}$
**Output:** $\mathbf{X}^*$ converging to a local minimum of $E_{\text{total}}$; Deformed source model $\{\mathcal{V}^*, \mathcal{E}\}$;
         Correspondence $\Pi^*_{\mathcal{ST}}, \Pi^*_{\mathcal{TS}}$ between $\mathcal{S}$ and $\mathcal{T}$.

---

**1**   Initialize deformation graph $\mathcal{DG}$ and $\mathbf{X}^{(0)}$ with identity transformations; $F_{\mathcal{T}} = \mathbf{F}(\mathcal{T})$; k = 0;
**2**   **while** True **do**
**3**      Update source vertex $\mathcal{V}^{(k)}$ by Eqn.(7);
**4**      **if** $k\%100 == 0$ *and Flag == Stage-I* **then**
**5**         $F_{\mathcal{S}}^{(k)} = \mathbf{F}(\mathcal{V}^{(k)})$; $\Pi_{\mathcal{ST}}^{(k)} = \mathbf{NN}(F_{\mathcal{S}}^{(k)}, F_{\mathcal{T}})$; $\Pi_{\mathcal{TS}}^{(k)} = \mathbf{NN}(F_{\mathcal{T}}, F_{\mathcal{S}}^{(k)})$;
**6**      **if** $k\%100 == 0$ *and Flag == Stage-II* **then**
**7**         $\Pi_{\mathcal{ST}}^{(k)} = \mathbf{NN}(\mathcal{V}^{(k)}, \mathcal{T})$; $\Pi_{\mathcal{TS}}^{(k)} = \mathbf{NN}(\mathcal{T}, \mathcal{V}^{(k)})$;
**8**      Compute the set of filtered correspondences $\mathcal{C}^k$;
**9**      $\mathbf{X}^{(k+1)} = \arg\min E_{\text{total}}$ by Eqn.(12)
**10**     **if** converged and Flag == Stage-I **then** Flag = Stage-II;
**11**     **if** converged and Flag == Stage-II **then return** $\Pi^*_{\mathcal{ST}}$; $\Pi^*_{\mathcal{TS}}$; $\mathcal{V}^*$;
**12**     k = k + 1;

---

$$d_{h,l}(\mathbf{X}) = d_{h,l}(\Theta, \Delta) = R(\Theta_h)(g_l - g_h) + \Delta_k + g_k - (g_l + \Delta_l). \tag{11}$$

Here, $g \in R^{H \times 3}$ are the original positions of the nodes in the deformation graph $\mathcal{DG}$, and $\psi(h)$ is the 1-ring neighborhood of the $h-$th deformation node. $R(\cdot)$ is Rodrigues' rotation formula that computes a rotation matrix from an axis-angle representation and $\alpha$ is the weight of the smooth rotation regularization term.

**Total cost function:** The total cost function $E_{\text{total}}$ combines the above terms with the weighting factors $\lambda_{cd}, \lambda_{corr}$, and $\lambda_{arap}$ to balance them:

$$E_{\text{total}} = \lambda_{\text{cd}} E_{\text{cd}} + \lambda_{\text{corr}} E_{\text{corr}} + \lambda_{\text{arap}} E_{\text{arap}} \tag{12}$$

Now we are ready to describe our algorithm for minimizing $E_{\text{total}}$, which is shown in Alg. 1.

**Two-stage Registration:** Finally, we observe that solely depending on learned features to infer correspondence is sub-optimal. At the converging point, the deformed source shape is often at the right pose but has deficiencies in shape style. To this end, we perform a second-stage registration, based on the coordinates of deforming source and target. As shown in Sec. 4.2, such a design is clearly beneficial.

## 4 Experiments

**Datasets:** We evaluate our method and several state-of-the-art techniques for estimating correspondences between deformable shapes on an array of benchmarks as follows: **FAUST_r:** The remeshed version of FAUST dataset [4], which consists of 100 human shapes (10 individuals performing the same 10 actions). We split the first 80 as training shapes and the rest as testing shapes; **SCAPE_r:** The remeshed version of SCAPE dataset [2], which consists 71 human shapes (same individual in varying poses). We split the first 51 as training shapes and the rest as testing shapes; **SHREC19_r:** The remehsed version of SHREC19 dataset [36], which consists of 44 shapes of different identities and poses. We use it solely in test, and follow the test pair list provided by [36]; **SHREC07-H:** A subset of SHREC07 dataset [18], which consists of 20 heterogeneous human shapes of the varying number of vertices. We use it solely in test, and use the accompanied sparse landmark annotations to quantify all the pairwise maps among them; **DT4D-H:** A dataset proposed in [34], which consists of 10 categories of heterogeneous humanoid shapes. We use it solely in testing, and evaluating the inter-class maps split in [29]; **TOPKIDS:** This is a challenging dataset [25] consisting of 26 shapes of a kid in different poses, which manifest significant topological perturbations in meshes.

**Baselines:** We compare our method with an array of competitive baselines, including axiomatic shape registration methods: Smooth Shells [14], NDP [31], AMM [55]; learning-based registration

methods: 3D-CODED [19], Deep Shells [16], TransMatch [51], SyNoRiM [21]; deep functional maps frameworks: SURFMNet [45], WSupFMNet [46], GeomFMaps [13], DiffFMaps [35], NIE [24], ConsistFMaps [7], SSMSM [8]. According to input requirements, we put those relying on pure mesh input on the top, and the rest, suitable for point clouds, in the bottom of both tables.

**Train/Test Cases:** Throughout this section, we consider a highly challenging scenario: for each learning-based method, we train two models respectively with FAUST_r and SCAPE_r dataset, and then we run tests in a range of test cases including FAUST_r, SCAPE_r, SHREC19_r, SHREC07-H, DT4D-H and TOPKIDS. The test pairs of each case have been described above. In the following, $A/B$ means we train on dataset $A$ and test on dataset $B$.

**Datasets Alignment:** There exist extrinsically aligned versions for all the involved datasets but SHREC07-H. We equally feed in aligned datasets to all the baselines when available. On the other hand, the original version of SHREC07-H manifests significant variability of orientations. For the sake of fairness and simplicity, we apply our orientation regressor to it and provide baselines for our automatically aligned data. Note that, all the aligned datasets, as well as the synthetic dataset on which we train our orientation regressor, roughly share the same canonical orientation, which is defined by the SMPL [33] model. Finally, we always apply automatic alignment before inputting shapes into our pipeline, whatever the original orientation they are.

**Choice of Source Shape:** As mentioned at the beginning of Sec. 3, we compute correspondences between two raw input point clouds by associating them via a common source mesh $\mathcal{S}$. In Tab. 1 and Tab. 2, we indicate our choice of source mesh by Ours-*name*, where *name* indicates the origin of the source mesh. For simplicity, we fix the source mesh from each dataset and visualize them in the appendix. On the other hand, when implementing axiomatic shape registration methods, we only consider deforming the same template we pick from FAUST_r (resp. SCAPE_r) to every test point cloud in the test set of FAUST_r (resp. SCAPE_r), and establish correspondence by map composition, in the same manner as ours.

**Metric:** Though we primarily focus on matching point clouds, we adopt the commonly used geodesic error normalized by the square root of the total area of the mesh, to evaluate all methods, either for mesh or point cloud.

**Hyper-Parameters:** We remark that all the hyper-parameters are fixed for *all* experiments in this section. In particular, we settle them by performing a grid search with respect to the weights used in the final optimization to seek for the combination that leads to the the best registration results (quantitatively in terms of Chamfer distance and qualitatively by visual inspection) on a few training shapes. We provide more detailed discussion and ablation of the choice of hyper-parameters in the appendix.

## 4.1 Experimental Results

**Near-isometric Benchmarks:** As shown in Fig. 1, in the presence of large deformation between the template and the target, NDP[31] and AMM[55] fail completely while our method delivers high-quality deformation. Moreover, as illustrated in Table 1, our method achieves the best performance in 5 out of 6 test cases. Remarkably, in the two most challenging tasks, FAUST_r/SHREC19_r and FAUST_r/SHREC19_r, our method indeed outperforms *all* of the baselines, including the state-of-the-art methods that take meshes as input. Regarding point-based methods, SSMSM [8] performs well in the standard case and outperforms ours in FASUT_r/FAUST_r, but generalizes poorly to unseen shapes. Another important observation is that our method manifests robustness with respect to the choice of template shapes. In fact, the above observations remain true no matter which template we select.

**Non-isometric Benchmarks:** We stress test our method on challenging non-isometric datasets including SHREC07-H and DT4D-H. We emphasize that these test shapes are unseen during training.

1) SHREC07-H contains 20 heterogeneous human shapes, whose number of vertices ranges from 3000 to 15000. Moreover, there exists some topological noise (*e.g.*, the hands of the rightmost shape in Fig. 2 is attached to the thigh in mesh representation). As shown in Fig. 2, SSMSM [8] barely delivers reasonable results, which might be due to the sensitivity of graph Laplacian construction on point clouds. Topological noise, on the other hand, degrades mesh-based methods like Deep Shells [16]. Meanwhile, as shown in Tab. 2, our method achieves a performance improvement of

Table 1: Mean geodesic errors (×100) on FAUST_r, SCAPE_r and SHREC19_r. The **best** is highlighted.

| Method | Train | FAUST_r | | | SCAPE_r | | |
|---|---|---|---|---|---|---|---|
| | Test | FAUST_r | SCAPE_r | SHREC19_r | SCAPE_r | FAUST_r | SHREC19_r |
| Smooth Shells [14] | | 2.5 | \ | \ | 4.7 | \ | \ |
| SURFMNet(U) [45] | | 15.0 | 32.0 | \ | 12.0 | 32.0 | \ |
| NeuroMorph(U) [15] | | 8.5 | 29.0 | \ | 30.0 | 18.0 | \ |
| WSupFMNet(W) [46] | | 3.3 | 12.0 | \ | 7.3 | 6.2 | \ |
| GeomFMaps(S) [13] | mesh | 3.1 | 11.0 | 9.6 | 4.4 | 6.0 | 11.4 |
| Deep Shells(W) [16] | | 1.7 | 5.4 | 26.6 | 2.5 | 2.7 | 21.4 |
| ConsistFMaps(U) [7] | | **1.5** | 7.3 | 20.9 | **2.0** | 8.6 | 28.7 |
| AttentiveFMaps(U) [29] | | 1.9 | **2.6** | **6.2** | 2.2 | **2.2** | **9.3** |
| NDP [31] | | 20.4 | \ | \ | 16.2 | \ | \ |
| AMM [55] | | 14.2 | \ | \ | 13.1 | \ | \ |
| CorrNet3D(U) [59] | | 63.0 | 58.0 | \ | 58.0 | 63.0 | \ |
| 3D-CODED(S) [19] | | 2.5 | 31.0 | \ | 31.0 | 33.0 | \ |
| SyNoRiM(S) [21] | | 7.9 | 21.9 | \ | 9.5 | 24.6 | \ |
| TransMatch(S) [51] | pcd | 2.7 | 33.6 | 21.0 | 18.6 | 18.3 | 38.8 |
| DPC(S) [26] | | 11.1 | 17.5 | 31.0 | 17.3 | 11.2 | 28.7 |
| DiffFMaps(S) [35] | | 3.6 | 19.0 | 16.4 | 12.0 | 12.0 | 17.6 |
| NIE(W) [24] | | 5.5 | 15.0 | 15.1 | 11.0 | 8.7 | 15.6 |
| SSMSM(W) [8] | | **2.4** | 11.0 | 9.0 | 4.1 | 8.5 | 7.3 |
| Ours-SCAPE | | 3.4 | **5.1** | **5.4** | 2.6 | **4.0** | 5.1 |
| Ours-FAUST | | 3.0 | 6.3 | 5.9 | 2.9 | 4.0 | **4.8** |

Table 2: Mean geodesic errors (×100) on SHREC'07 and DT4D-H. The **best** is highlighted.

| Method | Train | FAUST_r | | SCAPE_r | |
|---|---|---|---|---|---|
| | Test | SHREC07-H(*) | DT4D-H | SHREC07-H(*) | DT4D-H |
| GeomFMaps [13] | | 30.5 | 38.5 | 28.9 | 28.6 |
| Deep Shells [16] | | 30.6 | 35.9 | 31.3 | 25.8 |
| ConsistFMaps [7] | mesh | 36.2 | 33.5 | 37.3 | 38.6 |
| AttentiveFMaps [29] | | **16.4** | **11.0** | **21.1** | **21.4** |
| TransMatch [51] | | 25.3 | 26.7 | 31.2 | 25.3 |
| DiffFMaps [35] | | 16.8 | 18.5 | 15.4 | 15.9 |
| NIE [24] | | 15.3 | 13.3 | 13.4 | 12.1 |
| SSMSM [8] | pcd | 42.2 | 11.8 | 37.7 | 8.0 |
| Ours-SCAPE | | 9.3 | 9.8 | **5.9** | 5.7 |
| Ours-FAUST | | **8.5** | \ | 6.1 | \ |
| Ours-CRYPTO | | \ | **6.9** | \ | **5.7** |

over 40% compared to the previous SOTA approaches (**8.5** vs. 15.3; **5.9** vs. 13.4), which again confirms the robustness of our approach. 2) Regarding DT4D-H, We follow the test setting of AttentiveFMaps [29], and only consider the more challenging inter-class mapping. Interestingly, our method outperforms the state-of-the-art point cloud-based approach by approximately 30% and exceeds the performance of mesh-based state-of-the-art methods by over 70%. Furthermore, even in the worst case, training on FAUST_r and using SCAPE template, our error is still the lowest compared to external baselines.

**Topologically Perturbed Benchmark:** The shapes in TOPKIDS present various adhesions – hand/arm adhered to abdomen/thigh (see Fig. 6 in the appendix). Given the ground-truth maps from each topologically perturbed shape to the reference shape, we evaluate all 25 maps (excluding the trivial one from reference to itself) with models trained on FAUST_r and SCAPE_r respectively. Since the orientation of the TOPKIDS dataset is not originally agreeable with the aligned version of the two training sets, we align the input with our orientation regressor and feed them into the baselines depending on extrinsic information [16; 35; 24; 8]. Regarding the template shape, we adopt the reference shape as our source mesh in registration. We report the quantitative comparisons in Tab. 3. It is obvious that topological noise poses great challenges for methods based on pure intrinsic information [13; 7; 29], while the counterparts perform relatively well. Especially, among the point-based methods, our method outperforms the latest SOTA method by a large margin, namely, over 40% relative error reduction (**7.1** vs. 12.3). We refer readers to the appendix for a qualitative comparison, which agrees with the quantitative results above.

Table 3: Mean geodesic errors (×100) on TOPKIDS which trained on FAUST_r and SCAPE_r. The **best** is highlighted.

| | GeomFMaps [13] | Deep Shells [16] | ConsistFMaps [7] | AttentiveFMaps [29] | DiffFMaps [35] | NIE [24] | SSMSM [8] | Ours |
|---|---|---|---|---|---|---|---|---|
| FAUST_r\TOPKIDS | 26.2 | **14.7** | 35.9 | 31.7 | 20.5 | 18.9 | 14.2 | **8.9** |
| SCAPE_r\TOPKIDS | 21.7 | **15.3** | 33.1 | 39.4 | 18.0 | 16.2 | 12.3 | **7.1** |

Table 4: Mean geodesic errors (×100) on different ablated settings, the models are all trained on SCAPE_r and test on SHREC'07.

| w/o Registration | w/o Stage I | w/o Stage II | w/o updating corre. | w/o cons. filter | Full |
|---|---|---|---|---|---|
| 11.5 | 10.6 | 10.1 | 8.1 | 7.2 | 5.9 |

Table 5: Mean geodesic errors (×100) of Ours, NDP, AMM based on the same initial maps.

| | SCAPE_r | SHREC19_r | SHREC07-H |
|---|---|---|---|
| Ini. | 5.5 | 8.1 | 11.5 |
| NDP | 5.4 | 11.4 | 8.9 |
| AMM | 11.4 | 10.7 | 8.8 |
| Ours | **2.6** | **5.1** | **5.9** |

## 4.2 Ablation Study

We report ablation studies in Tab. 4, in which we verify the effectiveness of each core design formulated in Sec. 3. Throughout this part, we train on SCAPE_r and test on SHREC07-H. In the registration stage, we use the SCAPE_r template as the source mesh and deform it to each point cloud from SHREC07-H. It is evident that each ablated module contributes to the final performance. In particular, both stages of registration play an equally important role in our framework.

Finally, to validate the superiority of our registration scheme, we report in Tab. 5 the average errors of the initial maps computed by our point-based DFM, and that of output maps of Ours, NDP [31], AMM [55], which are all based on the former. It is evident that, across three different test sets, our method consistently improves the initial maps, while NDP and AMM can even lead to deteriorated maps than the initial input.

## 5 Conclusion and Limitation

In this paper, we propose a novel learning-based shape registration framework without correspondence supervision. Our framework incorporates learning-based shape matching and optimization-based shape registration. Based on the former, our framework can perform registration between shapes undergoing significant intrinsic deformations; On the other hand, thanks to the latter, our method exhibit superior generalizability over the learning-based competitors. Apart from several designs tailored for our intuitive pipeline, we also introduce a data-driven solution to facilitate the burden of extrinsically aligning non-rigid points. We verify our framework through a series of challenging tests, in which it shows superior performance but also remarkable robustness.

**Limitation & Future Work** We identify two main limitations of our method: 1) Our optimization procedure involves iteratively updating correspondences between the intermediate point clouds and the target one, leaving sufficient room for improvement for efficiency; 2) Our pipeline is designed for full shapes, while it is applicable to partial targets, the registration quality is sub-optimal. In the future, we also plan to incorporate more existing techniques (*e.g.*, various regularization terms) from shape registration to further enhance the overall performance.

**Acknowledgement** The authors thank anonymous reviewers for their constructive comments and suggestions. This work was supported in part by the National Natural Science Foundation of China under contract No. 62171256, 62331006, in part by Shenzhen Key Laboratory of next-generation interactive media innovative technology (No. ZDSYS20210623092001004).

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

In this appendix, we provide more technical details and experimental results. In Sec. A, we provide 1) A detailed description of our orientation regressor in Sec. A.1; 2) More details on the registration method presented in Alg. 1 of the main text in Sec. A.3; 3) Implementation details of the whole pipeline in Sec. A.4.

In the second part, we provide more experimental results. We first provide more qualitative results in Sec. B.1, as well as quantitative results on training on large dataset and on animal dataset in Sec. B.2; Then we demonstrate the robustness of our method with respect to several perturbations in Sec. B.3; In Sec. B.4, we showcase the scalability of our method in matching real scans. Finally, we report a run-time decomposition and analysis in Sec. B.5.

Note that we also include a short video showcasing the registration process of our pipeline in the Supplemental Material.

# A Technical Details

## A.1 Orientation Regressor

Orientation plays an important role in point cloud processing. Classic point-based feature extractors [42; 43; 53] are not rotation-invariant/equivariant by design and, therefore are limited. A common practice is to leverage data augmentation [42]. However, such can only robustify the model under small perturbations, while falling short of handling arbitrary rotations in $SO(3)$. On the other hand, recent advances in designing rotation-invariant/equivariant feature extractor [41; 10] seem to shed light on alleviating alignment requirements in non-rigid point matching. However, we empirically observe that such a design generally degrades the expressive power of the feature extractor, leading to inferior performance of the trained DFM.

This issue also attracts attention from the community of shape matching. In [11], the rotation prediction, instead of directly estimating the transformation, is formulated as a classification problem. However, this method can only address the rotation around a fixed axis and exhibits relatively low accuracy. Other works [56; 57] transform the input point cloud into handcrafted features. However, it should be noted that this approach often increases the computational complexity due to the additional feature extraction and processing steps involved.

In this paper, we propose a data-driven solution to this problem, accompanied by a registration module that is robust with the estimated alignment (see Tab.3 in the main text). As mentioned in the main text, we adopt the regressor model proposed in [9]. We use the synthetic SURREAL shapes [52], which are implicitly aligned by the corresponding generative codes to train our orientation regressor. We follow [9] and use PointNet++ [43] as the feature extractor. As we assume to be given full shape, either mesh or point cloud, for each input cloud, we translate it so its mass lies at the origin point. Then we estimate the orientation deviation, $R$, of it from the orientation implicitly defined by the SURREAL dataset. Finally, we apply the inverse rotation $R^T$ on the input to obtain the aligned shape. We emphasize again that all the shapes, either train or test, go through this alignment in our pipeline.

## A.2 Deep Functional Maps (DFM)

We illustrate the DFM, both for meshes and point clouds in Fig. 4.

## A.3 Registration Algorithm

We provide a more detailed version of our registration algorithm in Alg. 2. given a source mesh $\mathcal{S}$, target point cloud $\mathcal{T}$ and a trained point feature extractor $\mathbf{F}$, we first initialize the deformation graph $\mathcal{DG}$ with QSlim method [17] and set $\mathbf{X}^{(0)}$ with identity transformations. After that, we compute the target point feature $F_{\mathcal{T}}$. During registration, we first deform the source model with a deformation graph, then we update the correspondence by computing the nearest neighbor between the source feature and target feature in stage-I or the source point cloud and target point cloud in stage-II. After we compute the set of filtered correspondences $\mathcal{C}^k$, we update the total cost function $E_{\text{total}}$ with the ADMM optimizer. Finally, we follow [31] to implement an early stop strategy in both stages.

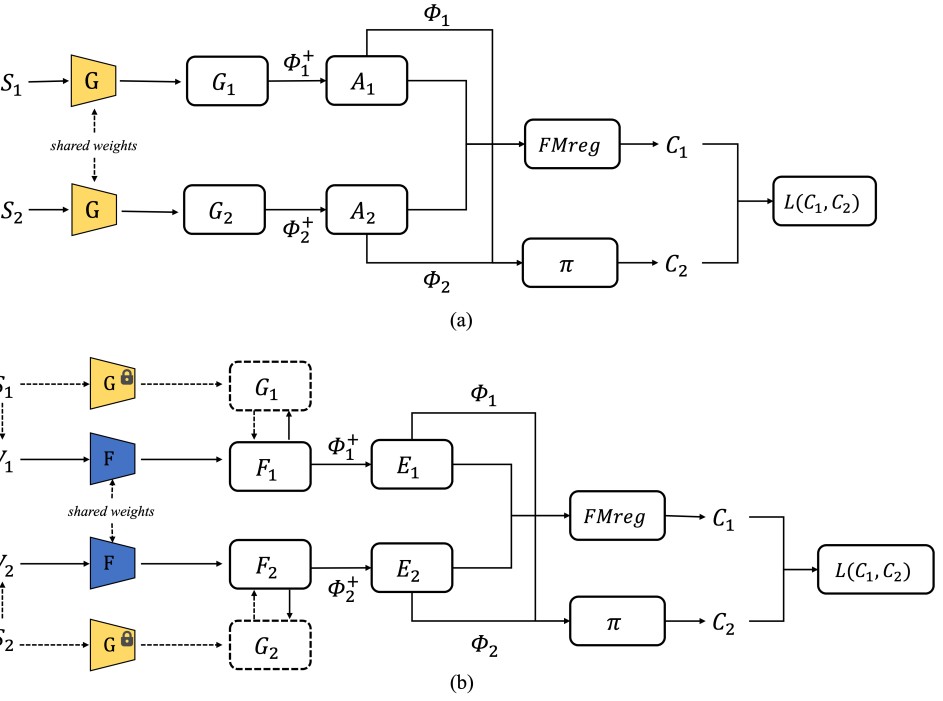

Figure 4: (a) DFM architecture for pre-training on meshes; (b) Our proposed architecture for training a point-based DFM, which is self-supervised by its mesh-based counterpart.

---

**Algorithm 2:** Shape registration pipeline.

---

**Input:** Source mesh $\mathcal{S} = \{\mathcal{V}, \mathcal{E}\}$ and target point cloud $\mathcal{T}$;Trained point feature extractor $\mathbf{F}$
**Output:** $\mathbf{X}^*$ converging to a local minimum of $E_{\text{total}}$; Deformed source model $\{\mathcal{V}^*, \mathcal{E}\}$;
  Correspondence $\Pi^*_{\mathcal{ST}}, \Pi^*_{\mathcal{TS}}$ between $\mathcal{S}$ and $\mathcal{T}$.

**1** Initialize deformation graph $\mathcal{DG}$ and $\mathbf{X}^{(0)}$ with identity transformations; $F_{\mathcal{T}} = \mathbf{F}(\mathcal{T})$;k = 0;
**2** $eps = 10^{-8}$; $count = 0$;
**3 while** True **do**
**4** $\quad$ $\mathcal{V}^k = \mathcal{DG}(\mathbf{X}^{k-1}, \mathcal{V}^{k-1})$;
**5** $\quad$ **if** $k\%100 == 0$ *and Flag == Stage-I* **then**
**6** $\quad\quad$ $F_{\mathcal{S}}^{(k)} = \mathbf{F}(\mathcal{V}^{(k)})$; $\Pi_{\mathcal{ST}}^{(k)} = \mathbf{NN}(F_{\mathcal{S}}^{(k)}, F_{\mathcal{T}})$ ; $\Pi_{\mathcal{TS}}^{(k)} = \mathbf{NN}(F_{\mathcal{T}}, F_{\mathcal{S}}^{(k)})$;
**7** $\quad$ **if** $k\%100 == 0$ *and Flag == Stage-II* **then**
**8** $\quad\quad$ $\Pi_{\mathcal{ST}}^{(k)} = \mathbf{NN}(\mathcal{V}^{(k)}, \mathcal{T})$ ; $\Pi_{\mathcal{TS}}^{(k)} = \mathbf{NN}(\mathcal{T}, \mathcal{V}^{(k)})$;
**9** $\quad$ Compute the set of filtered correspondences $\mathcal{C}^k$ ;
**10** $\quad$ $E_{\text{total}}^{(k)} = \lambda_{\text{cd}} E_{\text{cd}}(\mathcal{V}^{(k)}, \mathcal{T}) + \lambda_{\text{corr}} E_{\text{corr}}(\mathcal{V}_{\mathcal{C}^k}^{(k)}, \mathcal{T}_{\mathcal{C}^k}) + \lambda_{\text{arap}} E_{\text{arap}}(\mathcal{DG})$;
**11** $\quad$ $\mathbf{X}^{(k)} = \arg\min E_{\text{total}}^{(k)}$;
**12** $\quad$ **if** $(E_{total}^{(k)} - E_{total}^{(k-1)}) < eps$ **then**
**13** $\quad\quad$ count = count + 1;
**14** $\quad\quad$ **if** *count > 15* **then**
**15** $\quad\quad\quad$ **if** Flag == Stage-I **then** Flag = Stage-II; count = 0; eps = $10^{-7}$;
**16** $\quad\quad\quad$ **if** Flag == Stage-II **then** return $\Pi^*_{\mathcal{ST}}$;$\Pi^*_{\mathcal{TS}}$; $\mathcal{V}^*$;
**17** $\quad$ k = k + 1;

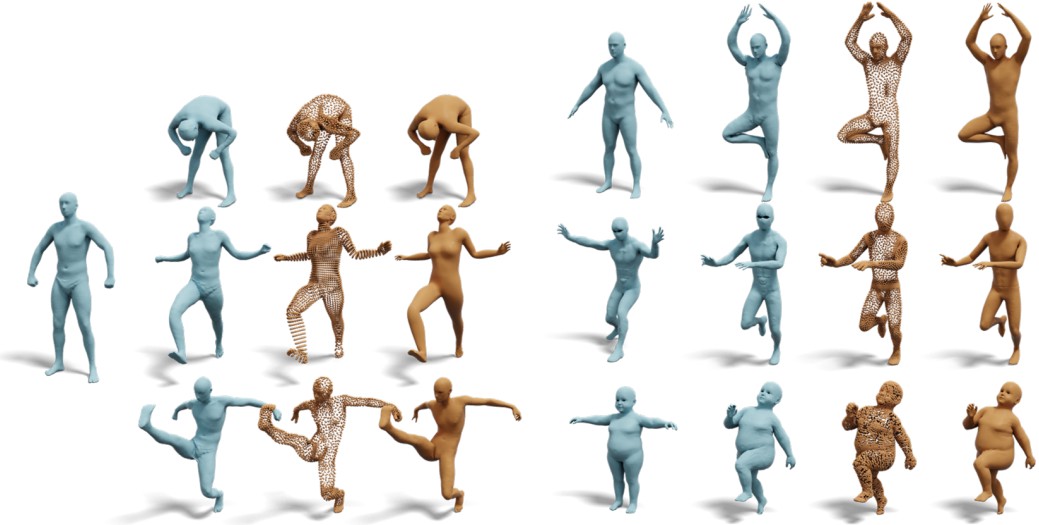

Figure 5: More non-rigid shape registration results. The original source mesh and its deformed counterpart are represented in blue, while the target point cloud and the ground truth mesh are depicted in orange.

## A.4 Implementation Details

We implement our framework in PyTorch. We use modified DGCNN [24] the backbone of our feature extractor. The output feature dimension is set to 128. In the training related to deep functional maps, we always use the first 50 eigenfunctions of Laplace-Beltrami operators. For training the DFM network, in Eqn.(6) of the main text, we empirically set $\lambda_{\text{bij}} = 1.0$, $\lambda_{\text{orth}} = 1.0$, $\lambda_{\text{align}} = $ 1e-4, $\lambda_{\text{NCE}}$ = 1.0. We train our feature extractor with the Adam optimizer with a learning rate equal to 2e-3. The batch size is chosen to be 4 for all datasets.

Regarding the registration optimization, in Eqn.(12) of the main text, we empirically set $\lambda_{\text{cd}} = 0.01$, $\lambda_{\text{corr}} = 1.0$, $\lambda_{\text{arap}} = 20$ in Stage-I and $\lambda_{\text{cd}} = 1.0$, $\lambda_{\text{corr}} = 0.01$, $\lambda_{\text{arap}} = 1$ in Stage-II. For $\alpha$ in Eqn.(9) of the main text, we set $\alpha = 0.2$. Note that the roles of $\lambda_{\text{cd}}$ and $\lambda_{\text{corr}}$ are reversed from Stage-I to Stage-II. In fact, in Stage-I, the initial shapes are likely to differ by large deformation, therefore it is more reliable to estimate correspondences via the learned embeddings. Once Stage-I converges, we empirically observe that the deformed source shape admits a similar pose with the target (see the video in Supplemental Material), therefore it is reasonable to further deform the source by extrinsic embeddings, *i.e.*, Chamfer distance on the coordinates. We stress that both losses are critical in both stages and perform the following ablation experiment on setting $\lambda_{\text{cd}}$ and $\lambda_{\text{corr}}$ to be zero independently in the two stages. Tab. 6 reports the scores. We find that most of the time, turning off leads to worse results, especially when the test shapes are heterogeneous (see, e.g., SHREC07-H).

Table 6: Mean geodesic errors (×100) of Ours on setting $\lambda_{cd}$ and $\lambda_{corr}$ to be zero independently in the two stages

|  | SCAPE_r | SHREC19_r | SHREC07-H |
|---|---|---|---|
| $\lambda_{cd} = 0$ in Stage-I | 2.5 | 5.5 | 6.9 |
| $\lambda_{corr} = 0$ in Stage-II | 2.7 | 5.6 | 6.8 |
| Ours | 2.6 | 5.1 | 5.9 |

# B Experiments

## B.1 Further Qualitative Results

**Visualization of Registration Results** In this section, we show more qualitative results of registration regarding different templates and different test shapes in Fig 5. Our method is capable of registering shapes with significant deformations across different templates.

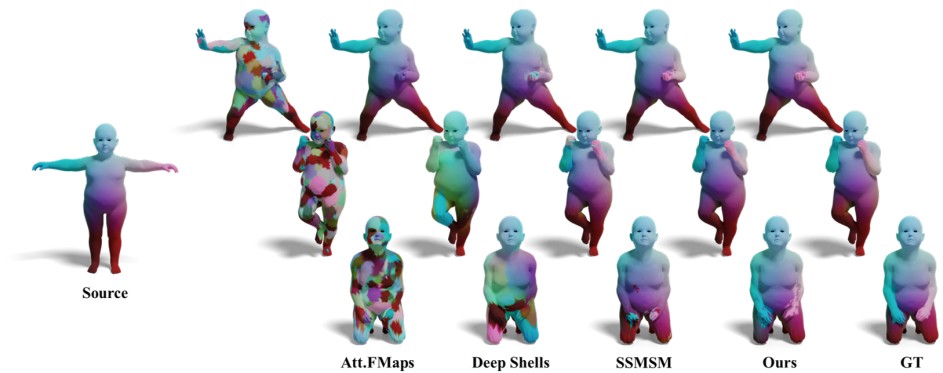

**Source**       **Att.FMaps**    **Deep Shells**    **SSMSM**    **Ours**      **GT**

Figure 6: We present the qualitative results of testing the TOPKIDS trained on SCAPE_r. Our method demonstrates robustness in regions of mesh topology with noise, maintaining stable performance.

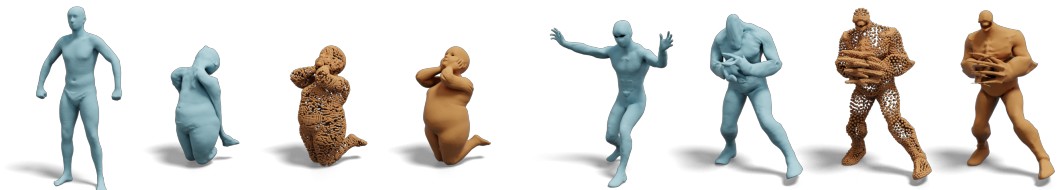

Figure 7: We showcase some examples of failed registrations.

**Qualitative Comparison in TOPKIDS:** In Fig. 6, we qualitatively visualize maps obtained by different methods tested in the TOPKIDS benchmark. It is obvious that our results outperform all the competing methods. Especially, a purely intrinsic method, AttentiveFMaps seriously fails due to the topological perturbation within the mesh structure.

**Failure Cases:** On the other hand, we also identify some failure cases, which are shown in Fig. 7. The two cases both suffer prominent heterogeneity between the template and the target. In the left case, where an ideal reference is given, we adapt it in registration instead of enforcing the hard highly non-isometric deformation with templates from SCAPE_r or FAUST_r (see Sec. B.3). While in the right case, it is not obvious how to choose an easier template for establishing correspondences between the two shapes, we, therefore, settle down at the setting described in Sec.4.2. It is worth noting that, though, we can easily identify that the problematic regions, *i.e.,* head, and hands from the visualization, which is unavailable for methods only return maps.

**Templates:** In the Fig. 8 we visualize the templates used in different experiments. We select a pose close to the T-pose from the training sets of SCAPE_r, DT4D-H, and FAUST_r, respectively, as our templates.

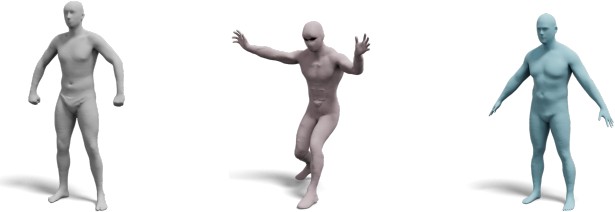

Figure 8: We respectively use template from SCAPE (left), CRYPTO (middle), and FAUST (right) in our experiments.

### B.2 Further Quantitative Results

**Trained on Large-Scale Dataset:** We have followed the setting of SSMSM [8] to train our feature extractor on the SURREAL 5k dataset [f] and to test it on the FAUST_r, SCAPE_r, and SHREC19_r datasets. We select one training shape from SURREAL 5k as the template in registration. As shown in Table 7, our method outperforms the competing non-rigid point cloud matching methods by a noticeable margin. We also refer the readers to Table 3 in [8] for a complete table including methods utilizing mesh input, and we highlight that our method is even comparable with the latter. In particular, ours vs. SOTA of mesh-based techniques over the three test datasets: 3.2 vs. 2.3, 2.5 vs. 3.3, 4.6 vs. 4.7. That is, we achieved the best overall result in two of the three test datasets.

Table 7: Mean geodesic errors (×100) trained on SURREAL

| Method | Train | SURREAL | | | \|Data\| |
|---|---|---|---|---|---|
| | Test | FAUST_r | SCAPE_r | SHREC19_r | |
| DiffFMaps | | 26.5 | 34.8 | 42.2 | 230k |
| DPC | | 13.4 | 15.8 | 17.4 | 230k |
| SSMSM | | 3.5 | 3.8 | 6.6 | 5k |
| Ours-SURREAL | | **3.2** | **2.5** | **4.6** | 5k |

**Trained on Animal Dataset:** We also perform experiments on the remeshed SMAL dataset. We first randomly generate 5000 shapes with SMAL model to train the alignment module. Then we train a DFM with the remeshed SMAL dataset. The template shape is 'dog_06' from the training set. The quantitative results are reported in Table 8. Remarkably, our method achieves more than a 40% performance improvement than the second-best baselines.

Table 8: Mean geodesic errors (×100) on SMAL_r.

| | TransMatch | DPC | DiffFMaps | NIE | SSMSM | Ours |
|---|---|---|---|---|---|---|
| SMAL_r | 20.7 | 21 | 17.1 | 16.3 | 7.2 | **4.2** |

### B.3 Robustness

**Robustness with respect to Input Perturbation:** We evaluate the robustness of different methods with respect to noise and orientation perturbations in Tab. 9. In the former, we perturb the input point clouds with Gaussian noise. In the latter, we randomly rotate the input point clouds within ±30 degrees along $x, y, z-$axis. Thanks to our optimization-based approach and orientation regressor, in both tests we achieved the best accuracy, but also the lowest standard deviation over three runs of experiments.

**Robustness with respect to Missing Parts:** In Tab.1 of the main text, we report shape matching results on SHREC19_r dataset, where the $40^{th}$ shape is commonly excluded in the test phase of prior works [29; 8]. The main reason is that this shape is incomplete, consisting of two main connected components (see the abdomen of the source shape in Fig. 9). Therefore, for intrinsic methods depending on mesh eigenbasis, such disconnectedness makes spectral alignment extremely difficult, if not impossible. On the other hand, extrinsic methods show a more robust outcome. In particular, we show in Fig. 9 that our framework can reasonably deform the template mesh (from SCAPE_r) to

Table 9: Mean geodesic errors (×100) on under different perturbations. Noisy PC means the input point clouds are perturbed by Gaussian noise. Rotated PC means the input point clouds are randomly rotated within ±30 degrees along $x, y, z-$axises respectively. The standard deviation value is shown in parentheses.

| | Unperturbed PC | Noisy PC | Rotated PC |
|---|---|---|---|
| DiffFMaps [35] | 12.0 | 14.9(2.57) | 26.5(3.35) |
| NIE [24] | 11.0 | 11.5(0.32) | 19.9(1.29) |
| SSMSM [8] | 4.1 | 5.4(0.11) | 9.2(1.01) |
| Ours | **2.6** | **2.9(0.06)** | **3.6(0.96)** |

the incomplete point cloud with a significantly isolated portion, which leads to superior matching results compared to the baselines.

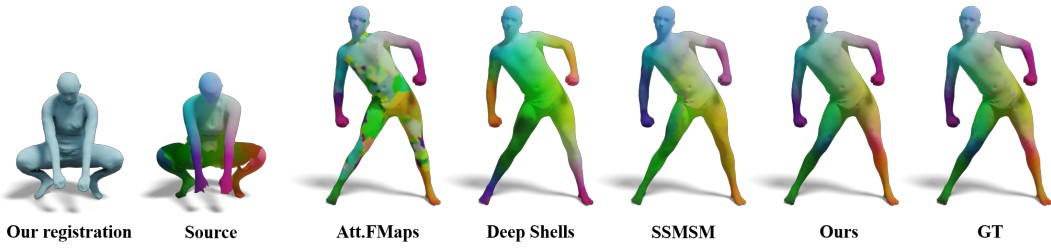

Figure 9: A challenging case from SHREC19_r dataset, see the text for details.

**Robustness of Orientation Regressor:** To demonstrate that our learned orientation regressor (see Sec.3.1 of the main text), we visualize its outcome on shapes from SHREC07-H, which manifests notable diversity in sampling density, size, and initial orientation.

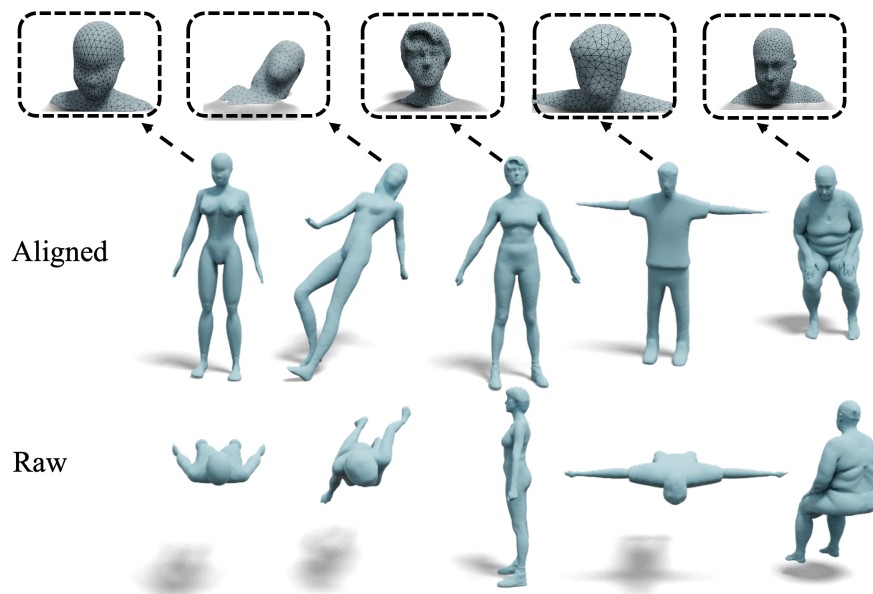

Figure 10: Bottom row: The input shapes of SHREC'07 exhibit various orientations; Top row: The shapes aligned by our orientation regressor. Note that there exists significant differences in sampling density.

## B.4 Scalability

Our approach is scalable w.r.t input size. Thanks to the fact that we non-rigidly align shapes in $\mathbb{R}^3$, we can in theory freely down- and up-sample both the template mesh and the target point clouds. Note this is non-trivial for methods based on mesh or graph Laplacian, as deducing dense maps with landmark correspondences over graph structures is a difficult task on its own. In Fig. 11, we show the matching results on the real scans from the FAUST challenge, each of which consists of around 160,000 vertices. In contrast, [8] can handle at most 50,000 vertices without running out of 32G memory on a V100 GPU. We visualize its matching results on the subsampled (to 40,000 vertices) point clouds for comparison.

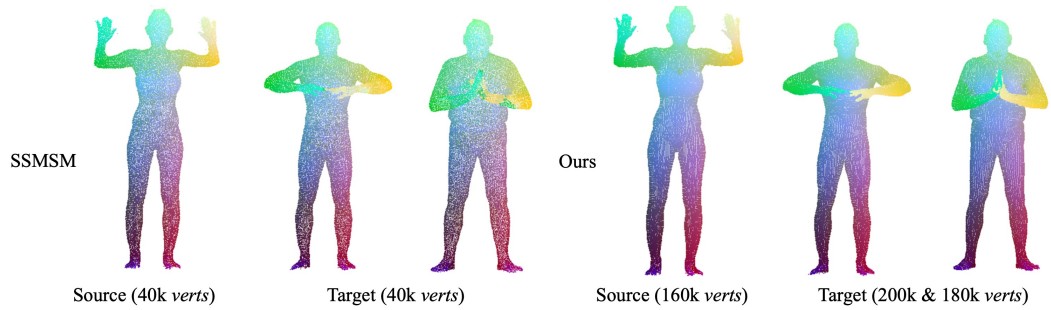

SSMSM        Ours

Source (40k *verts*)    Target (40k *verts*)     Source (160k *verts*)    Target (200k & 180k *verts*)

Figure 11: Qualitative results on matching MPI-FAUST raw scan data. Our method can match the point clouds at the original resolution, while SSMSM [8] can not.

## B.5   Run-time Analysis

In this part, we analyze the time cost of our optimization-based registration module. In particular, we separate it into two main parts, initialization, and iterative optimization. For the former, we in general fix a source mesh as a template, and then we construct a deformation graph on it as described in Sec.3.2 in the main text. Since the deformation graph can be pre-computed and cached for further use of the same template, we put more emphasis on the latter.

In particular, we perform registration between the template shape from SCAPE_r and the 20 test shapes therein and compute the average run-time of each core step in the optimization. As illustrated in Tab. 10, our method achieves an average optimization time of approximately 13.4 seconds per shape, with about 90% of the total time being dedicated to Stage-I. Analysis depicted in Fig. 12 reveals that the most time-consuming operation is the ADMM backward step, while the combined time cost of our correspondence filtering and updating operations accounts for less than 3% of the overall execution time. Though the reported run-time is not quite satisfying, we emphasize that to evaluate maps across $n$ point clouds, we only need to register them to the same template shape ($O(n)$), which is different from the pair-wise ($O(n^2)$) mapping and post-processing approaches [7].

Table 10: Average time and iteration cost for each shape.

|  | Total | Stage-I | Stage-II |
|---|---|---|---|
| Time cost(s) | 13.4 | 12.0 | 1.4 |
| Iterations | 1274. | 1130. | 144. |

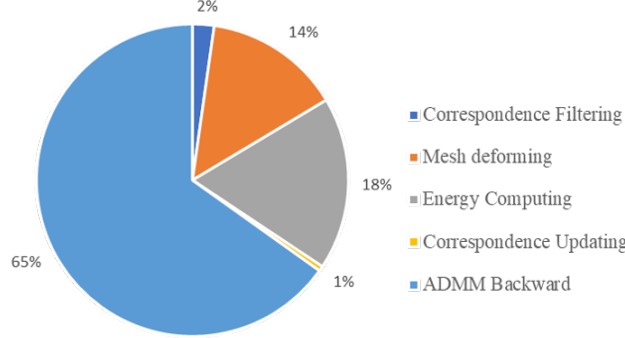

Figure 12: Run-time decomposition of our registration module.

