# OpenReview forum: "Non-Rigid Shape Registration via Deep Functional Maps Prior"
_NeurIPS.cc/2023/Conference — NeurIPS 2023 poster_

### Official Review · Reviewer_GUi9 · 2023-06-19

**Soundness:** 3 good
**Presentation:** 4 excellent
**Contribution:** 3 good
**Rating:** 5
**Confidence:** 4

**Summary:**

The method proposes an unsupervised pipeline to solve 3D shape-to-shape registration. Firstly, the two shapes are aligned by a pre-trained orientation regressor. Then, a soft-correspondence is obtained by a Point Feature extractor, optimized using a Deep Functional Maps schema, and considering several unsupervised regularizations (e.g, bijectivity, orthogonality). Finally, the source shape is iteratively deformed using a Deformation graph (using Chamfer distance, the learned correspondence, and as-rigid-as-possible), and the correspondence is updated every 100 iterations. The method is tested on humanoid datasets (FAUST_r, SCAPE_r, SHREC19_r, SHREC07-H, DT4D-H),  and extensively compared against many other approaches, both  supervised and unsupervised. obtaining interesting results.

== FINAL RATING ==

In the discussion phase, authors provided much further evidence and clarified my concerns. I notice, however, that while a majority is leaning toward acceptance, there is not a consensus among the reviewers. Going through the reviewers that assigned negative scores, I do not find significant ground to change my rating. In particular, Reviewer xFDL mainly criticizes the novelty of the work but does not point to other works to support this claim. While I can understand that the work does not seem particular novel in its component (and the general principle of registration + correspondence in feature embedding can be found in previous works (e.g., more SmoothShell, DeepShell than DPC, while both of them are designed for meshes), I do not find itself a proper ground of rejection (as also reported in the Reviewing Guidelines; also combinations of existing techniques are valuable). The experimental evaluation is appreciable, the performance convincing, and I do not see this as incremental w.r.t. any other previous work. I find this an interesting contribution to a research field that counts only a limited amount of work.
Instead, I see the detailed review and discussion of Reviewer iB8H, and I think it contains many valuable observations that could improve the paper. However, summarizing the reported criticisms seems to be:

A) novelty w.r.t to DPC; which in the last comment looks solved, or at least significantly tuned down

B) general motivation of the work; while I see that the underlying principle and positioning of the work in literature can be improved, overall, I do not see other works that perform similarly, and I struggle to see this as a "follow-up" of a specific methodology to consider this an incremental contribution of something in particular. Also, the obtained results seem to be already a reasonable justification for the proposed approach (since I think we all agree that the paper effort is beyond engineering work, and hence, they communicate a promising research direction)

C) Other details (e.g., clarifying the role of hyper-parameters, missing citations, rephrasing); that I think make sense, but they can be easily addressed in the camera ready, and I do not consider a sufficient ground for rejection.

For these reasons, I lean toward acceptance. I suggest authors incorporate the suggestions (especially about suggested experiments and paper positioning in literature), and I wish them the best of luck with their work.

**Strengths:**

1) Not many methods are available to solve shape-to-shape correspondence in an unsupervised way; the proposed approach smartly combines existing techniques and well-established methods to obtain good results.

2) The proposed approach outperforms existing direct competitors, even some supervise approaches. I am sure it would be of interest for the shape-matching community.

3) The paper is well presented, clear and direct, I enjoyed the reading, and I had no problem understanding the main components. I am sure the method can be re-implemented with limited effort.

**Weaknesses:**

1) The general applicative context of the method is unclear. From the introduction and the pipeline figure, I had the impression that the method is designed to obtain correspondence between point cloud. However, the method is about registering a mesh to a point cloud, which can be extended to the point cloud case, using the source mesh as a bridge. This, however, is not properly tested: while other approaches also show noisy point clouds with different levels of noise (e.g., [30]), in this case, only point clouds sampled from meshes are reported, with a gaussian noise that is not detailed nor in the main manuscript or in the supp.mat. . Given the nature of the used feature extractor (which I expect to be quite sensitive to real noise and clutter), I suggest including more details on the considered Gaussian noise, and test on real raw pointclouds (which could also be another failure case, but would point the reader to interesting future directions)

2) While I appreciate the simplicity of the proposed approach and the combination, I struggle to see a general message or insight. Also, I do not see the method as directly applicable since it is trained only on a few shapes, and we do not know how it would scale on larger datasets (e.g., AMASS), and I guess it would fail to solve registrations in real contexts (e.g., noisy scans), since it would ruin the learning on the underlying surface. It is also tested only on a domain (humans) with a significant amount of labelled data. So, without these elements, I wonder if the paper could be of significant impact or if it is just a carefully designed pipeline that will be just another matching method without much influence in the field. I suggest discussing the main message, analysing the scalability of the approach, and testing on domains in which labelling data are much more complicated to obtain (e.g., chairs, which also would make stronger the claim about topological noise in the supplementary)

**Questions:**

1) In Table 1 some results are not reported. Why?
2) The orientation regressor module resembles the input transform module of PointNet. Wouldn't a rotation augmentation for the feature extractor work (and remove the need for a further module)? Another possible alternative is given by the method proposed in [A].
3) Given that overall the pipeline assumes some degrees of bijectivity, how would it perform in the presence of a significantly different number of vertices between the two shapes (e.g., by a factor of x100)? Even considering two complete shapes which have a bijectivity on the surface, the losses are defined vertex wise, and might lead to a degradation of the regularization impact.

Minor:
1) I see that reference [30] is incorrect in the bibliography, and [31] has names compressed. I suggest double-checking the references.
2) Relevant references not discussed: [B], [C], [D]

[A]: Adjoint Rigid Transform Network: Task-conditioned Alignment of 3D Shapes, Zhou et al., 3DV 2022 \
[B]: NeuroMorph: Unsupervised Shape Interpolation and Correspondence in One Go, Eisenberger et al., ECCV 2021 \
[C]: NCP: Neural Correspondence Prior for Effective Unsupervised Shape Matching, Attiki et al., NeurIPS 2022 \
[D]: 3d-coded: 3d correspondences by deep deformation, Groueix et al., ECCV 2018

**Limitations:**

Limitations are discussed in the main paper.

---

> ### Author Rebuttal · Authors · 2023-08-09
>
> Thank you for the constructive comments and the recognition of our contributions. Below we address the comments:
>
> **Applicative context of the proposed method:**
> We actually follow the same scheme of adding noise perturbation as DiffFmaps [30].
> We highlight our new results reported in Rebuttal Mat., which demonstrate that our method can be *directly* applied to matching large-scale real scans, as well as be extended to perform matching and registration regarding partial, even noisy point clouds.
>
> **Scalability in terms of training set:**
> First of all, we consider the ability of our method to efficiently and effectively learn from a small-scale training set as an advantage. That being said, we have followed the setting of SSMSM [8] to train our feature extractor on the SURREAL 5k dataset [f] and to test it on the FAUST_r, SCAPE_r, and SHREC19_r datasets. We select one training shape from SURREAL 5k as the template in registration. As shown in Table 2 of the rebuttal material, our method outperforms the competing non-rigid point cloud matching methods by a noticeable margin. We also refer the readers to Table 3 in [8] for a complete table including methods utilizing mesh input, and we highlight that our method is even comparable with the latter. In particular, ours vs. SOTA of mesh-based techniques over the three test datasets: 3.2 vs. 2.3, **2.5** vs. 3.3, **4.6** vs. 4.7. That is, we achieve the best *overall* result in two of the three test datasets.
>
> **Extension to more challenging datasets, such as chairs:**
> We highlight that our unsupervised matching network is essentially trained with the prior that the underlying maps among training shapes are isometric and bijective. Especially, the former plays a pivotal role in the development of DFM frameworks. In the case of chairs, such prior can be violated significantly (e.g., chairs can be with or without arms, back, etc.), making direct extension challenging. We believe that exploring effective matching priors for such data is an interesting future direction to investigate.
>
> **Missing scores in Table 1:**
> There are three axiomatic (optimization-based) methods in Table 1 -- Smooth Shells [12], NDP [27], and AMM [45]. Obviously, they do not have generalization results. Regarding the rest missing scores, we apologize for not being able to perform all the tests upon the deadline. On the other hand, we would like to point out that the regarding methods do not perform strongly in the accomplished, typically simpler, tests, which therefore has little impact on the overall comparison and analysis.
>
> **Alternatives to orientation regressor:**
> We emphasize that our current pipeline can in general take arbitrarily rotated point clouds as input (See Fig. 3 in the Rebuttal Mat.), performing data augmentation in SO(3) can be quite heavy, and might not be optimal (see Table 1, 2 in [g] for reference).
>
> **Dependence on bijectivity:**
> First of all, one simple solution is to down-sample the target point clouds. Note that our pipeline is primarily designed for matching organic shapes like humans and animals (see the latter in the rebuttal material), down-sampling the underlying smooth surfaces in general would not lose many high-frequency signals. Once the template mesh is deformed to fit the down-sampled point clouds, then one can easily infer the maps regarding the full-resolution point cloud, since all the shapes are explicitly, non-rigidly aligned.
>
> In practice, we also observe the robustness of our pipeline regarding the number of vertices. For instance, the number of vertices of the shapes in SHREC07 ranges from 2,000 to 16,000, while the template shape has around 5,000 points. And our method achieves decent registration results (see Fig. 1) and outperforms the baselines by a large margin (see Table 2) under such perturbation, beyond the heterogeneity.
>
> **References issues:**
> We thank you for pointing out the relevant prior works and would be happy to include and discuss them in the future revision. Of course, we will fix the typos in the current references as well.
>
> [f] Learning from synthetic humans, G. Varol, et al., CVPR 2017.
> [g] Vector Neurons: A General Framework for SO(3)-Equivariant Networks, C. Deng, et al., ICCV 2021.

---

> > ### Comment · Reviewer_GUi9 · 2023-08-13
> > **Post-Rebuttal**
> >
> > I thank the authors for their reply to my concerns.
> >
> > I appreciate the amount of experiments and different settings provided in the rebuttal.
> >
> > I would just say that probably the dependence on bijectivity can be elaborated on in the limitation section. This is because even if it is true that "down-sampling the underlying smooth surfaces, in general, would not lose many high-frequency signals", ideally, some applications would obtain the highest possible precision. Matching could have more "scales" (i.e., we seek a global but also a local coherence), and at higher frequencies, the details are (often) more prone to vary across different instances of the same class (and bijectivity does not hold much). Of course, it is not a request to solve also this, and I will not consider this a weakness, but more a discussion point to enhance the conclusion/limitation discussion.
> >
> > I do not have other questions, and I maintain my positive opinion. Looking forward to hear from other reviewers their feedback.

---

> > > ### Author Response · Authors · 2023-08-15
> > >
> > > We thank you for the reply and positive opinion. Regarding the problem of bijectivity dependency, we think that the challenging cases you mentioned are beyond the scope of the current submission and that it can be indeed an interesting future direction (e.g., matching high-resolution human faces with the proposed pipeline). We will add the respective discussion in the future revision.

---

### Official Review · Reviewer_CNuX · 2023-06-28

**Soundness:** 3 good
**Presentation:** 4 excellent
**Contribution:** 2 fair
**Rating:** 5
**Confidence:** 5

**Summary:**

In this paper an unsupervised non-rigid shape registration method is proposed. The proposed method combines intrinsic spectral mapping (i.e. based on the deep functional map framework) together with  extrinsic deformable shape registration (i.e. a deformation graph) to enable unsupervised 3D deformable shape matching. In many challenging benchmark datasets, the proposed method demonstrates competitive matching performance, better cross-dataset generalisation ability, and robustness against noise and rotation of input shapes.

**Strengths:**

1. The paper is well-written and easy to follow. The main contribution and methodology are well illustrated.
2. The paper integrates shape matching and shape registration into the same framework. The shape matching part is based on the deep functional map framework to obtain point-wise correspondences. The shape registration part is based on the deformation graph to non-rigid align two shapes to refine the final correspondences.
3. In order to enable matching shapes with different orientation, the paper proposes an orientation regressor to align shapes into a canonical frame.

**Weaknesses:**

1. The novelty of the method is limited. The proposed method consists of three components (rotation regressor, feature extractor, shape registration) and each component is derived from prior works without large modifications.
2. The interconnection of the components in the method is missing. The first stage is to train a rotation regressor to align shapes into the canonical frame. The second stage is to train a feature extractor to obtain correspondences. The third stage is to optimise the deformation graph to align two shapes. Every stage is somehow separated during the training stage, while it is desired to see the connection of shape matching and shape registration like Deep Shells or NeuroMorph.

**Questions:**

1. Since the proposed method is based on test-time optimisation, what is the convergence speed of the proposed method?
2. Since the deep functional map can be fully intrinsic (i.e. with intrinsic input features), what if we use it to obtain the initial correspondences and rigidly align the shapes into the canonical frame without using rotation regressor?


**Limitations:**

1. The proposed method is tailored to complete shape matching, so it cannot achieve desirable matching results for partial shapes.
2. The proposed method is based on iterative optimisation to align two shapes, so the runtime is slower than other learning-based methods.

---

> ### Author Rebuttal · Authors · 2023-08-09
>
> Thank you for the constructive comments and the recognition of our contributions. Below we address the comments:
>
> **Modules are separated in the pipeline and lack connection:**
> Thank you for the constructive comment. We agree that integrating shape matching and shape registration in a more associated manner can be desirable. However, we would like to highlight that Deep Shells and NeuroMorph fully leverage the structural information within the meshes. More specifically, Deep Shells extensively computes the first 500 Laplacian eigenpairs, and NeuroMorph uses mesh connectivity in graph neural network training and dense geodesic distance matrices on both input shapes for training loss. Such intrinsic geometric information is critical for the respective non-rigid shape registration. In contrast, our method expects nothing but raw point clouds during inference (the template mesh is fixed), which can be challenging for extracting intrinsic information (see, e.g., shapes in Fig. 3 of the supplemental material).
>
> To overcome this difficulty, we propose to learn a teacher network (DiffusionNet) on a small number of training meshes (80 in all of our experiments), and then train a student network (DGCNN) that consumes points but infers intrinsic features by mimicking the teacher network. Finally, the student network is frozen and used to estimate correspondences dynamically during the final registration part. If we were to integrate shape matching and registration as an associated pipeline, it would make more sense to put the teacher network, the student networks, as well as the registration component all trainable. However, it would violate our main goal -- performing shape registration on raw point clouds.
>
> As a proof of concept, we have tried to unfreeze and fine-tune the point feature extractor during the registration process on the SCAPE_r dataset, in the following two ways: 1) Updating point-wise correspondences per 100 iterations as in Alg. 1, leading to a slight performance drop (from 2.6 to 2.8); 2) Updating per iteration, leading to a failure of convergence.
>
> **Convergence speed:**
> We have reported the average convergence steps as well as a full running-time decomposition in our Supp. Mat. (see Table 2 and Fig.6 therein). Especially, our method converges within 1274 iterations (1130+144) on average, tested in the Scape test set.
>
> **Use intrinsic methods to obtain rigid alignment:**
> We emphasize again that our main target is to perform *directly* shape matching/registration on raw point clouds. In particular, our pipeline, once trained, can deform a given template mesh to target point clouds without any pre-processing on the latter. Note that pre-processing may be slow and parameter sensitive, and is beyond the scope of this paper.
>
> **Applicability on partial point clouds:**
> In Fig. 1 and Fig. 2 in the Rebuttal Mat., we demonstrate some preliminary results on extending our pipeline to matching partial, even noisy point clouds. Essentially, we train a DFM tailored for partial-view point clouds generated on SCAPE_r dataset, and replace the two-way Chamfer distance with a one-way one. Note that in this experiment we assume the partial point clouds are rigidly aligned. Nevertheless, we believe the results have sufficiently shown the potential of our general scheme.

---

> > ### Comment · Reviewer_CNuX · 2023-08-18
> >
> > I thank the authors for the clarifications. I will keep my initial rating.

---

### Official Review · Reviewer_1VPz · 2023-07-05

**Soundness:** 3 good
**Presentation:** 2 fair
**Contribution:** 3 good
**Rating:** 7
**Confidence:** 3

**Summary:**

The paper describes a method for corresponding a 3D triangle mesh to a point cloud of a similar (possibly articulated) shape.  The two-stage process first corresponds the two shapes in a high-dimensional feature space, then corresponds them again using geometric features while deforming the source closer to the target.

**Strengths:**

The approach makes sense, and the results seem strong compared to previous work, both qualitatively and quantitatively.

**Weaknesses:**

In general, the presentation could be improved.

There should be a figure analogous to Figure 1, but showing qualitative correspondences for cases where train/test datasets do match. Only showing severe failures of competing methods on very disparate shapes does not convey the full picture.

The supplementary video showing the progress of the correspondence algorithm is very helpful. It would be even more informative if the mesh was texture mapped with the checkerboard as in Figure 1.

Equations, Figures, Tables need more thorough descriptions.

  Equation 1:
          Even though the text says that one can optimize the feature function F when it introduces equation 1, it is unclear how the function F is optimized. If not including an equation involving F, you could at least point to a specific equation number in [11].
  Equation 2: Parameter alpha should be defined after this equation, rather than after equation 4.
  Equation 3: What does the cross symbol represent?
  Equation 4: Parameter n2 is not used in the expression. Should the denominator summation go up to n2?
  Table 1 & 2:
          What are the two sections separated by a horizontal line (they each have a bold set of numbers)?
          Train/Test column contains no values.
  Table 2:  What is "Ours-CRYPTO", this acronym is not discussed/referenced in the text?
  Table 3: What is "Ideal PC"?

Algorithm should also initialize "Flag" (presumably to Stage-I).

**Questions:**

Could this approach be implemented without learned functions? The intro mentions that spectral mapping can be geometric (rather than learned), and the second stage is akin to non-rigid ICP.

Would it make sense to run Stage-I again after Stage-II, and repeat both stages a few times?

**Limitations:**

The method requires a triangle mesh for the source shape, which is not discussed in the limitations section. The exposition would be stronger if the authors took the source shape as a point cloud and ran some automatic meshing on it for their experiments.

Another limitation that the authors mention is that the method only works on "full shapes". I assume this means that the whole source and target surfaces are adequately sampled. It would help if the authors suggested some ideas on how one would go about lifting this requirement.

---

> ### Author Rebuttal · Authors · 2023-08-09
>
> Thank you for the constructive comments and the recognition of our contributions. Below we address the comments:
>
> **Improve paper presentation:**
> Thank you for the suggestions on improving the presentation of the paper, we would be happy to incorporate all of them in the future revision. Regarding Fig. 1, our intension is to illustrate the heterogeneity presented in the SHREC07 dataset, as well as our stronger generalization capacity than the competing baselines. The more thorough evaluation results are reported in Table 2, which also agree with the qualitative ones in Fig. 1. Below are responses to the minor comments:
>
> 1. We would be happy to clarify Eqn. (1)-(4) in the future revision. On the other hand, we would like to refer the readers to Sec. 1 of the Supp. Mat. for a more self-contained description;
>
> 2. The cross symbol (\dagger) in Eqn. (3) indicates the pseudo inverse of a matrix;
>
> 3. The PointInfoNCE loss is introduced to enforce the output features of DGCNN (for point clouds) to be *point-wisely* close to that of DiffusionNet (for mesh). The features are computed on the same shape (without and with mesh connectivity, respectively). A similar loss regarding S_2 is used as well;
>
> 4. In Table 1 and 2, the methods above the horizontal lines are *designed to* take meshes as input. The rest methods take point clouds directly;
>
> 5. Ours-CRYPTO in Table 2 indicates that a shape belongs to the category ‘Crypto’ of the DT4D-H dataset is used as template;
>
> 6. Ideal PC in Table 3 means the clean, aligned point clouds.
>
> **Implement our approach with non-learned functions:**
> We highlight that our pipeline requires no pre-processing or pre-alignment on the input point clouds. It is possible to implement with axiomatic spectral method. However, it would require to build graph Laplacian on-the-fly, since during registration the template is dynamically deformed. Such approach can be less efficient, and may suffer from scalability issues.
>
> **Repeat Stage-I and -II for multiple times:**
> Thank you for the suggestion. We have performed the whole registration procedure twice on the Scape dataset. The registration error decreased from 2.57 to 2.49, i.e., an improvement of 3.1%. It is worth noting, though, such approach on average introduces an over 60% computational overhead (convergence steps: 1274 vs. 1931).
>
> **Perform meshing on the source point cloud:**
> Thank you for the suggestion for improving the utility of our approach, which we believe is surely feasible. We think exploring it in a principled way can help to lift the need of template mesh in the future.
>
> **Applicability on partial point clouds:**
> In Fig. 1 and Fig. 2 in the Rebuttal Mat., we demonstrate some preliminary results on extending our pipeline to matching partial, even noisy point clouds. Essentially, we train a DFM tailored for partial-view point clouds generated on SCAPE_r dataset, and replace the two-way Chamfer distance with a one-way one. Note that in this experiment we assume the partial point clouds are rigidly aligned. Nevertheless, we believe the results have sufficiently shown the potential of our general scheme.

---

> > ### Comment · Reviewer_1VPz · 2023-08-17
> > **Thanks for the rebuttal**
> >
> > My rating remains.

---

> > > ### Author Response · Authors · 2023-08-18
> > >
> > > Thank you for the reply and the positive feedback to our work.

---

### Official Review · Reviewer_xFDL · 2023-07-06

**Soundness:** 2 fair
**Presentation:** 2 fair
**Contribution:** 2 fair
**Rating:** 4
**Confidence:** 3

**Summary:**

This paper proposes an unsupervised framework for non-rigid shape registration. The proposed method deforms source mesh towards the target point cloud, guided by correspondences induced by high-dimensional embeddings learned from deep functional maps. Empirical results show that the proposed method achieves state-of-the-art results on several benchmarks for non-rigid point cloud matching.

**Strengths:**

1. The overall writing is fluent;

2. The proposed method outperforms state-of-the-art approaches on several benchmarks;

**Weaknesses:**

1. The organization can be further improved (the organization of Methodology does not follow Fig. 1), which makes the paper sometimes hard to follow (probably due to the complexity introduced by assembling 3 stages for tackling the problem);

2. The novelty of this paper would be a major concern. It seems the proposed algorithm in this paper simply assembles three stages, each with an existing method. The orientation regressor is from [9]. The feature extractor is the modified DGCNN proposed in [22]. The non-rigid registration mainly follows [18].

3. The submission and main paper have different titles. Please fix it.


**Questions:**

1. Why is the proposed method unsupervised? It seems the orientation regressor requires the ground-truth pose to the canonical space. And the PointInFoNCE loss (Eq. 4) also requires correspondence labels? Or the authors call the generalization from one dataset to the other datasets the unsupervised learning?

2. How good is the orientation regressor? Can it deal with large translations? e.g., the translation introduced by a walking human. From my experiences, this kind of pose regressor can only overfit to a specific model and hard to generalize. Also, it is usually not very accurate (correct me if I was wrong);

3. Why all the models are called "pre-trained"? Should it be generalizing a model trained on the same task to new data?

**Limitations:**

Limitations have been discussed in the main paper.

---

> ### Author Rebuttal · Authors · 2023-08-09
>
> Thank you for all the insightful comments. First of all, we would be happy to improve the presentation as suggested, as well as to fix the inconsistent titles in the future revision. Below we address the comments:
>
> **Novelty:** We acknowledge that many components in our pipeline are inspired by existing works. However, we would like to emphasize that the way they are integrated are novel, which leads to a simple yet effective solution to the challenging problem of matching **unstructured** point clouds undergoing significant deformations. Beyond that, to the best of our knowledge, our solution to train a point-based feature extractor that respects intrinsic geometry, but without ground-truth correspondence labels is novel. We kindly refer the readers to the Author Rebuttal for a more detailed response.
>
> **Why is the method unsupervised?**  A more precise description of our supervision could be **correspondences label-free**. Given the opportunity, we would be happy to clarify this in the future revision. We acknowledge that the orientation regressor indeed relies on the ground-truth pose, which is obtained by simply fixing certain parameters in the SMPL generative model. On the other hand, we emphasize the fact that our method requires **no** ground-truth correspondence, either sparse or dense, throughout the pipeline. Obviously correspondence labels are much more difficult to obtain in practice.
>
> **PointInfoNCE requires labels:** The PointInfoNCE loss is introduced to enforce the output features of DGCNN (for point clouds) to be *point-wisely* close to that of DiffusionNet (for mesh). The features are computed on the same shape (without and with mesh connectivity, respectively). Therefore, we simply use the identity map between the same set of vertices (regarded as mesh vertices and point clouds respectively) on each shape to formulate Eqn. (4), which does *not* require any non-trivial correspondence label.
>
> **Performance of orientation regressor and its robustness regarding translation:** Since we assume the completeness of input point clouds in the submission, deviations induced by large translations can be resolved by moving the mass centers of all point clouds to a fixed point, which is also a standard practice in shape registration.
>
> Regarding the quality of orientation regressor, as shown in Fig. 3 of the Rebuttal Mat., though it is trained on a set of generated shapes sharing the same mesh connectivity, it can handle heterogenous shapes of varying number of vertices (2,000~20,000) well and delivers reasonable rigid alignments.
>
> Moreover, we empirically observe that our registration component enjoys certain robustness regarding the imperfect initial orientation. In particular, we *turn off* the orientation regressor, and *directly* perform registration on the rotated point clouds (see experiments reported in Table 3 of the main submission) and get **4.7(1.03)**/mean(std) in mean geodesic errors. New results show that our method still produces better and more stable results than the baselines. We attribute such robustness to the fact that non-rigid shape registration also involves deformations with respect to extrinsic orientation deviations.
>
> **Why all models are called "pre-trained"?** We call the orientation regressor and the point-based DGCNN *pre-trained* to emphasize that our key module, the final registration component, is optimization-based, leading to a geometrically meaningful procedure (see, for example, the video clip in the Supp. Mat.). The learning-based models are pre-trained and frozen during registration.

---

> > ### Comment · Reviewer_xFDL · 2023-08-17
> > **Reviewer Feedback**
> >
> > Thanks for the feedback! That helps me understand your work better. As pointed by all the reviewers, the major flaw of this paper is the limited novelty. I think this is the most important part in evaluating a paper, and based on that, I will keep my initial rating, although it is a little bit negative.
> >
> > Moreover, I hope the authors could carefully revise their paper afterwards, as there are some places in the current version unclear and misleading.
> >
> > Cheers!

---

> > > ### Author Response · Authors · 2023-08-18
> > >
> > > Thank you for the reply, and we are glad that our rebuttal helps you to understand our framework better. We would be happy to revise our paper for a better and clearer presentation.
> > >
> > > **Novelty discussion:**
> > >
> > > We have actually devoted length to the discussion of our main contributions and novelty in the Author Rebuttal (see at the top of the page) and would be more than happy to address your *further comments/questions* regarding it.
> > >
> > > Our key novelties are two-fold, as re-iterated below for ease of discussion:
> > >
> > > 1. To the best of our knowledge, we are the first to address the problem of matching **unstructured** point clouds undergoing significant deformation via a hybrid approach. The relevant prior works depend on either heavily mesh structure (NeuroMorph, Deep Shells), or correspondence supervision (TransMatch). In contrast, our method can deform a template to a raw point cloud in a *direct and unsupervised* manner.
> > >
> > > 2. We propose a novel self-supervised learning scheme to infer intrinsic-aware features from unstructured point clouds effectively and efficiently. Compared with the relevant and concurrent work [8], our design is more general (not adhered to DiffusionNet architecture, not based on graph Laplacian construction) and flexible (can be extended to more powerful and/or more tailored-for backbones).

---

### Official Review · Reviewer_iB8H · 2023-07-07

**Soundness:** 2 fair
**Presentation:** 2 fair
**Contribution:** 2 fair
**Rating:** 4
**Confidence:** 5

**Summary:**

This paper studies the problem of non-rigid shape matching. The problem is first decomposed into learning (rigid) orientation of shapes and   then learning shape matching on aligned shapes. For the later, it proposes to learn it with a  sequential pipeline, consisting of various modules, that is also optimized in a two stage manner.  the main idea is to facilitate learning by similarity in both ambient space (R^3) as well as high dimensional learned feature space. To this end, it trains a DGCNN feature extractor based on combination of several loss functions (DFM prior loss, ARAP loss, Chamfer/cosine loss). The method is validated on  isometric as well non isometric benchmarks.

Post Rebuttal:

The rebuttal provided additional experiments to support some of its claim as mentioned below. I lean towards rejection since the work is a sequential concatenation of existing individual modules (also noted by reviewer CNuX and xFDL ) without any conceptual justification to why those modules need to be combined e.g. why do we need to combine a DFM and ARAP loss. The rebuttal also contains several unsubstantiated statements detailed below. Combining two lines of research is not a technical contribution or contribution to community if there no conceptual justification given. Majority of the introduction section needs to be rewritten and contextualised *correctly* wrt prior work. Besides, submission had several hyper-parameters whose values were missing and there is no mention of model sensitivity towards these hyper-parameters even though the approach is unsupervised.

**Strengths:**

- Experimental results on various near isometric benchmark look promising (Table 1) and compares extensively with existing work.

- the paper identifies the generalization problem of some embedding based approach [30,8] to unseen shape data.

**Weaknesses:**

- Presentation: Section 1 contains several factually incorrect or unsupported claims. Moreover, the motivation of the work is misplaced since their is no validation of it later in the experimental section. Please support all the intuitive claims either with a prior reference or by explicitly mentioning results from this paper.

a)  why do we need to decompose the shape matching problem into learning an alignment first and then matching aligned shapes? Is it
 because [8], [22] do so or is there a scientific justification behind it as shown empirically in rigid shape matching or non-rigid shape
 matching with DFM. e.g. it is shown to provide an extrinsic supervision that helps to disambiguate symmetry issues in DFM. Please motivate the problem/solution accordingly.

b)  Line 46 < prior work with resulting high dimensional embedding lack intuitive geometric meaning> Since this is used as one of 3
reasons behind this paper/formulation, please demonstrate this on an example where in contrast, this paper obtains an intuitive
embedding with geometric meaning.

c)  Line 54-57: Please provide a reference to support these claim or prove it later with visualization and examples in experimental section.

d) [30] requires shapes to be pre-aligned at train or test time. This is not true. You can train and test [30] without such alignment and it makes no such assumption on input requirements in the paper[30].


-  Novelty: the main conceptual idea/key insight (Line 52) in this work is to enforce similarity in both ambient space as well as learned feature space. DPC [b] proposed the exact same idea with DGCNN, cosine/chamfer loss in the most simplest possible way for non-rigid shape matching. The idea is already well known that others have even built upon it for non-rigid shape matching e.g. [a]. So claiming this as a key insight is not justified IMO.

-  Missing references and comparison of a very similar work[b]:  authors should also compare their work with DPC on their benchmarks. this will show what gains (if any) are brought by DFM prior or ARAP loss.

a. Learning Canonical Embeddings for Unsupervised Shape Correspondence with Locally Linear Transformations
b. DPC: Unsupervised deep point correspondence via cross and self construction, 3DV 2021.

-  formulation without a principled approach: the network consisting of (DGCNN, DFM prior, ARAP loss, chamfer loss, pointinfo loss etc) combines modules from different frameworks without any principled reason. e.g. DFM prior from DFM literature, DGCNN feature extractor, chamfer & cosine loss from DPC etc. in a two stage optimization procedure. why do we need a DFM prior when we are deforming a source and a target shape with a ARAP loss ?

-  Too many hyperparameters in an unsupervised approach : The paper should mention in the main body how many hyperparameters  overall does this approach has and how were their values chosen? I count more than 10. Moreover, how is it justified to choose different values for the same hyperparameter  (weighing scalars) in an algorithm and call the resulting approach *unsupervised*?  there is a two order magnitude of difference in hyperparameter values ($\lambda_{cd}$ and $\lambda_{corr}$) between different runs of the algorithm (Stage 1 and Stage 2).

**Questions:**

- for Non-isometric, SMAL is considered the main benchmark and there is already a large literature that benchmark their results on SMAL. why ignore such standard benchmark?

**Limitations:**

yes

---

> ### Author Rebuttal · Authors · 2023-08-09
>
> We thank you for the constructive comments on our motivation and novelty. Below we address the comments.
>
> **Novelty, especially compared with DPC:** We argue that our approach is *fundamentally* different from DPC in the following perspectives:
>
> 1.	Our feature extractor for point clouds is learned in an intrinsic-aware way, while such information is absent in learning the counterpart of DPC;
>
> 2.	Our registration component explicitly deforms the template towards the target point cloud. In DPC, the cross-reconstruction is essentially a re-indexing of the target point cloud via soft maps induced by the latent proximities. For instance, Eqn. (3) of DPC amounts to a point permutation when the weights deduced in Eqn. (2) approximate a delta distribution;
>
> 3.	Finally, like DFM-based approaches [8, 22], DPC infers correspondences via high-dimensional embeddings, while our approach does so in the ambient space, which is more intuitive and easier to analyze.
>
> Moreover, we compare our method with DPC on the near-isometric benchmarks. As shown in Table 1 & 2 in the Rebuttal Mat., our method outperforms DPC by a *significant* margin when trained on both small-scale and large-scale datasets. We refer the readers to Table 3 of [8] for more detailed results.
>
> **Principle of the proposed formulation:** We refer the readers to the Author rebuttal for detailed motivation of our framework design.
>
> **Motivation of alignment and matching:** We emphasize that our approach performs extrinsic shape registration on *raw* point clouds, in a direct fashion without any explicit mesh/graph construction. It is then clear that our approach is sensitive with respect to position and orientation of the input. Many relevant works either implicitly (NDP [27], AMM [45]) or explicitly (SSMSM [8], NIE [22]) require rigid aligned shapes as input/initialization. Another way out is to leverage dense correspondence labels (DiffFMaps [30]). We remark the latter is not by construction robust w.r.t SO(3) perturbations. As shown in Table 3 of our main submission, it is sensitive w.r.t rotations when trained on aligned point clouds, without the effective data augmentation in the original implementation.
>
> In contrast, we leverage the generative model and propose a principled solution to this challenge, which lifts the rigid alignment in inference or dense correspondence labels in training.
>
> **Geometric meaningful embeddings:** We clarify that our high-dimensional embeddings, similar to that of [8, 22], are obtained by an uninterpretable learned network, which also lacks geometric meaning. In general, correspondences induced by such embeddings are difficult to evaluate and analyze without ground-truth maps. In contrast, our formulation leverages the learned embeddings to deform a template shape explicitly towards a given target point cloud, which provides a more geometrically intuitive mapping/registration procedure. We particularly refer the readers to the video in the Supp. Mat. As a result, one can perform both qualitative (by visually comparing the deformed template and the target) and quantitative (by Chamfer distance or RMSE) analysis on the output maps, even without ground-truth maps.
>
> **Claim made in Line 54-57:** The regarding claim has been justified in Fig. 3 and Table 1 of the main submission. Especially, we compare our method with NDP [27] and AMM [45], which depend on proximities in the ambient space to iteratively estimate correspondences and can fail significantly in the presence of large deformation. In Fig. 3, we deform the FAUST template (the 3rd shape from left) to the right-most shape. It is obvious that ambient proximities would lead to erroneous maps in the beginning, making it difficult to guide the right deformation (to rise arms significantly). The quantitative results in Table 1 also validate this observation. Our method achieves at least 78% matching error reduction compared to them on the two standard benchmarks.
>
> **Motivation of using DFM prior when ARAP loss is used:** The ARAP loss in general serves as a regularizer for shape registration, which prevents the deformed template from being overly distorted. Solely using ARAP loss does not lead to successful registration, as it could not provide any cue for matching. We refer the readers to Table 4 of our main submission: Comparing *w/o Stage-I* and *Full*, we can see the DFM prior (i.e., Stage-I) contributes significantly.
>
> **Too many hyper-parameters:** The hyper-parameters used in the pre-trained models (orientation regressor and DFM) follows the regarding prior works. As for those for the registration optimization, we search for the optimal hyper-parameters with *2 pairs of training shapes* of SCAPE w.r.t the registration loss, which does not depend on any correspondence label. We remark that our hyper-parameters are fixed across different template meshes, training sets, and test sets
> .
> **Experiments on SMAL:** As requested, we have performed experiments on the *remeshed* SMAL dataset [d]. We first randomly generate 5000 shapes with SMAL model [e] to train the alignment module. Then we train a DFM with the remeshed SMAL dataset [d]. The template shape is ‘dog_06’ from the training set. The quantitative results are reported in Table. 3 of the Rebuttal Mat. Remarkably, our method achieves more than a 40% performance improvement than the second-best baselines.
>
> [d] Complex functional maps: a conformal link between tangent bundles, N. Donati, E. Corman, S. Melzi, M. Ovsjanikov, CGF 2022. [e] https://smal.is.tue.mpg.de

---

> > ### Comment · Reviewer_iB8H · 2023-08-14
> > **follow up on rebuttal**
> >
> > thank you for your time and effort. The reviewer has gone through the supplement and main submission multiple times as suggested in rebuttal. Comments below: \
> >
> > - Algorithm 1 (pseudo-code) not same in Main submission and supplement:
> >  There is an additional early stopping count condition in Line14-17 of Algorithm 1 in supplement. no such thing exist in algorithm 1 of main submission. I assume the results shown in main submission are based on the algorithm from supplement? if so, please explain what this early stopping (count <15) criteria is. why do we need it especially if the algorithm has another stop criteria (max_iteration =100) in the underlying optimization. (Line 183 in main submission). why $E_i -E_{i-1} < eps$ would not suffice for termination of any algorithm?
> >
> > \
> >
> > - Claims made in Line 54-57: The rebuttal repeatedly points to the table 1 and  texture transfer figure 1 to justify these intuitive claims.These table and figures contain the end result of this approach ( a concatenation of 5 full fledged siggraph/TOG/CVPR papers (DGCNN, DiffusionNET, orientation regresssor, modified DFM, ARAP)  along with geodesics to tackle a single problem).  the comparison with NDP/AMM  to justify these claims  is unfair since this submission takes a SOTA non rigid shape matching method (DiffusionNet+modified DFM) as a head start. a fair way to justify these intuitive claims of tackling large deformation with superiority over NDP/AMM or any other baseline would be to also initialize them with the same SOTA non rigid shape matching (e.g. w/o registration baseline in ablation table).
> >
> >  \
> >
> > - Too many hyperparameters and heuristics for an unsupervised approach:  In addition to the count, iteration parameters outlined above, the rebuttal/submission also misses out on critical hyper parameters and their values etc:
> >
> >     -- Threshold parameter in correspondence filtering and its value, how it is chosen
> >
> >     --  Rebuttal mentions $\lambda_{cd}, \lambda_{corr}, \lambda_{arap}$ were chosen based on 2 training shape pairs but does not mention how even though their values are critical to understand what contributes to overall performance in shape registration pipeline (details below). Since these parameters are different for different stages, they should also be indexed accordingly to distinguish better.
> >
> > \
> >
> >
> > - Stage 1 and Stage 2 in Shape registration: Based on the $\lambda_{cd}, \lambda_{corr}, \lambda_{arap}$ values in two different stages (and two order of magnitude difference between them), stage 1 does not rely on chamfer distance whereas stage 2 does not rely on correspondence filtering. Does it not imply the network absolutely needs geodesic based filtering in stage 1 ? and that chamfer distance based on feature similarity( & NN) is not effective/needed in Stage 1?   this is speculative since submission/rebuttal does not show the relative strength of different loss terms to gauge individual contribution within these individual stages.
> >
> > \
> >
> > -  alignment and matching: robustness is different than requirement of pre-aligned shapes. the submission should replace [30] with [38] (first work to show results with this requirement) or change the text accordingly.
> >
> > -  Geometric meaningful embeddings: please clarify the same in Introduction.

---

> > > ### Author Response · Authors · 2023-08-15
> > > **Responses to further comments**
> > >
> > > Thank you for the detailed reply. Below we address your new comments:
> > >
> > > **Inconsistencies in algorithm descriptions in the main submission and Supp. Mat.:**
> > >
> > > Due to the lack of space, we defer some algorithmic details to the more complete version in Supp. Mat. The algorithms are essentially the same.
> > >
> > > First, the 'converged' condition in Line 10, 11 of Alg. 1 in the main submission is described in details in Line 12-16 of the version in the Supp. Mat., which is the only stopping criteria in our algorithm. Second, Line 183 in the main submission is not an early stopping criteria. It says that we update point-wise correspondences between deforming template and the target with the learned embedding every 100 iterations during stage-I.
> > >
> > > **Ablation study on utlizing DFM output as initialization to NDP, AMM:**
> > >
> > > According to your suggestion, we have performed ablation studies to compare our pipeline with NDP and AMM based on the same initial correspondences. In particular, we train the DFM on the training set of SCAPE_r and use the SCAPE template (see Fig. 3 in the main submission) in all experiments.
> > >
> > > In the following table, we report the average errors of the initial maps computed by DFM, and that of output maps of Ours, NDP, AMM, which are all based on the same initial maps. It is evident that, across three different test sets, our method consistently improves the inital maps (at least 37% error reduction), while NDP and AMM can even lead to deteriorated maps than the initial input. These results shows the advantage of our proposed pipeline, especially the registration part.
> > >
> > > Test set     |      Ini.       |       Ours         |            NDP        |           AMM
> > >
> > > SCAPE_r     |      5.5     |       **2.6(-52%)**      |      5.4(-2%)     |      11.4(+107%)
> > >
> > > SHREC19_r  |     8.1     |       **5.1(-37%)**     |     11.4(+40%)   |    10.7(+32%)
> > >
> > > SHREC07-H  |    11.5     |     **5.9(-48%)**      |      8.9(-22%)     |     8.8(-23%)
> > >
> > > **Dissusion on hyper-parameters:**
> > >
> > > Thank you for the suggestions on clarifying the roles and choices of hyper-parameters. We would be happy to include a detailed discussion on it in the future revision.
> > >
> > > **How exactly the hyper-parameters (e.g., threshold in correspondence filtering, $\lambda_{cd}$, $\lambda_{corr}$, $\lambda_{arap}$) are chosen:**
> > >
> > > As we mentioned in the rebuttal, performing shape registration allows to evaluate the resulting registration/maps without ground-truth correspondences. In particular, we perform grid search with respect to the weights used in the final optimization to seek for the combination that leads to the the best registration results (quantitatively in terms of Chamfer distance and qualitatively by visual inspection). We seek for the threhold in correspondence filtering in a similar way, and it is set to 0.01 across all experiments.
> > > We emphasize again that (a) The above hyper-parameter selection is quite loose and may be *suboptimal*, since only two pairs of training shapes (in fact, the template source shape is the same) are involved; (b) We use the same hyper-parameters for *all* of our experiments.
> > >
> > > **The roles of $\lambda_{cd}$, $\lambda_{corr}$ at different stages:**
> > >
> > > Essentially, the registration procedures in stage-I and -II are respectively guided by proximities in the high-dimensional embedded space and the ambient space.  On the other hand, the loss $E_{corr}$ (measuring how the deformation agrees with the maps induced by the learned embeddings) and $E_{cd}$ (measuring the discrepency between the deformed template and the target in R^3) quantify them respectively. Therefore, it is natural to put more weight on the resepctive loss in the corresponding stage. The exact ratio, 100:1 and 1:100, is determined as described above.
> > >
> > > In fact, we have not tried to turn off either $E_{corr}$ or $E_{cd}$ in any stage before. This is motivated by our initial motivation -- to enforce the deformed shape to be close to the target in *both* the high-dimensional embedded space and the ambient space.
> > >
> > > According to your suggestion, we have also performed ablation study on setting $\lambda_{cd}$ and$\lambda_{corr}$ to be zero independently in the two stages. The following table reports the scores. We find that most of the time, turning off either leads to worse results, especially when the test shapes are heterongenous (see, e.g., SHREC07-H).
> > >
> > > Test set        |           Ours    |     $\lambda_{cd}$ = 0 in Stage-I   |     $\lambda_{corr}$ = 0 in Stage-II
> > >
> > > SCAPE_r       |     2.6               |              **2.5(-4%)**                     |              2.7(+4%)
> > >
> > > SHREC19_r     |   **5.1**             |               5.5(+8%)                    |               5.6(+10%)
> > >
> > > SHREC07-H    |   **5.9**              |              6.9(+17%)                   |              6.8(+15%)
> > >
> > > **Revising arguments on alignment & matching and geometrically meaningful embeddings:**
> > > We would be happy to revise these in the future version.

---

> > > > ### Comment · Reviewer_iB8H · 2023-08-18
> > > > **Final remarks based on reviews, rebuttal and follow up**
> > > >
> > > > Thank you for your answers. The rebuttal has clarified my concerns about unsupported claims in the submission with further experiments. However,  many of the new claims on contributions/novelty in rebuttal are handwavy , unconvincing or presented without justifications (details below) and therefore my doubt remains on the presentation of future/revised versions (unlike journal, a reviewer will not see the revised version?)
> > > >
> > > >  I revise my my final rating but keep it slightly negative, same as reviewer xFDL.
> > > >
> > > >  \\\\
> > > >
> > > > -  *Key conceptual idea contextualized wrt DPC*: As mentioned in the initial submission (Line 52), key is to enforce similarity in both ambient space as well as learned feature space. Closest work on matching unstructured point cloud is DPC with cross reconstruction/self reconstruction modules. While the cross reconstruction module is very similar conceptually in enforcing similarity via latent space, I agree self reconstruction is different than this work. In my opinion, submission should stick to this (Shape registration Stage 1 and 2 ) while revising future versions instead of the following:
> > > >
> > > >
> > > > - unconvincing/handwavy claims on contribution in rebuttal: \\\\
> > > >
> > > >
> > > >     a) *combining learning based and optimization-based shape matching*: the two terms are not exclusive : every single follow up paper on DFM is a learning based paper with underlying optimization (autodiff) of DL frameworks. The fact that this one uses an explicit ADMM based optimization is not a contribution.
> > > >
> > > >
> > > >   b) *first to address the problem of unstructured point cloud matching via a hybrid approach*: why do we need a hybrid approach when there are already existing frameworks for unstructured point cloud matching that does not require a mesh/laplacian. emphasis on 'first to do hybrid matching' is not convincing. the submission can argue that it is less rigid in requirement than some purely mesh based methods but it sill relies on DiffusionNet  to start with (underlying bottleneck).
> > > >
> > > >
> > > >   c) *generative modelling*, *lifting* alignment requirement etc: Such terms should be avoided since generative models in ML is a well defined term and this submission did not formalize a generative model in submission.
> > > > \\\\
> > > >
> > > > - Back to Shape registration Stage 1 and 2 and supporting its claim in future version:
> > > > \\\\
> > > >
> > > >    a) Contextualizing the key conceptual idea and citing DPC (as mentioned above).
> > > >
> > > >    b) Proper referencing and a motivation for alignment and matching.
> > > >
> > > >    c) Including rebuttal experiments to support claims made in line 54-57.
> > > >
> > > >    d) clarifying Geometric meaningful embeddings claim.
> > > >
> > > >    e) supporting Shape registration Stage 1 and 2  with an ablation study as done above.
> > > >
> > > >    f) clearly mention this approach introduces 8 hyperparameters (6 in stage 1/2+ corr. filtering threshold + early stoping count) and how critical they are.to overall performance and model sensitivity to them.
> > > >
> > > > In my view, above modifications are must to support the key idea and resulting claims.
> > > > \\\\
> > > >
> > > > - nice to have that will make submission stronger:
> > > >
> > > >   --why do we need DFM prior with ARAP loss: rebuttal mentions an empirical answer. a more convincing could be show if their output somehow are uncorrelated so combining them makes atleast a statistical sense if not geometric one.
> > > >
> > > > -- generalisation of this shape registration to other methods: reviewers understands it requires effort but it only makes the approach more effective if shape registration module/DFM prior can be taken and plugged in other methods and improves their performance as well.
> > > >
> > > > -- rename correspondence filtering to Geodesic filtering.
> > > >
> > > >
> > > > -- similarly rename feature extractor to something more specific that conveys student-teacher feature extractor

---

> > > > > ### Author Response · Authors · 2023-08-18
> > > > > **Clarification on novelty.**
> > > > >
> > > > > We sincerely appreciate your detailed comments and suggestions. We are glad that the rebuttal has clarified some of your concerns and would be happy to incorporate the revisions and experimental results in the future version.
> > > > >
> > > > > Below we mainly address your concerns about novelty and contributions (presented in our rebuttal):
> > > > >
> > > > > 1.    Our pipeline is based on two series of works on estimating correspondences between non-rigid shapes: 1) shape matching via learning a high-dimensional embedding (DFM-like works [8, 22, 30, 38], DPC); 2) non-rigid shape registration (NDP, AMM). To the best of our knowledge, the two lines have long been developed **in parallel**. We have discussed the respective limitations: 1) high-dim embeddings are obtained in an uninterpretable way, which is often *hard to analyze* and can *suffer from poor generalization*; 2) non-rigid shape registration methods are typically designed for non-rigidly aligning consecutive scans of the same instance, therefore they can fail when the shapes are of *large deformations or of distinctive styles*.
> > > > >
> > > > > 2.    Following (1), our **novel hybrid approach** combines these two lines of work. Rather than a random combination, we put them in a unified framework to compensate for each other’s limitations -- The learned embeddings provide more suitable proximities than the extrinsic coordinates in matching shapes, while by performing explicit non-rigid alignment we can introduce regularizations such as ARAP so that the alignment is more reasonable and plausible, which is easier to analyze and more reliable in matching/registrating unseen shapes.
> > > > >
> > > > > 3.    At this stage, it may seem trivial to put them together. We emphasize that we aim to deal with **unstructured point clouds**, which is the key challenge. Overall, the embeddings have to be computed on raw point clouds, ideally without mesh/graph construction (for both efficiency and robustness regarding topological noise). To this end, our novel teacher-student learning scheme is general, flexible, and effective.
> > > > >
> > > > > 4.    Finally, we highlight that the meshes and DiffusionNet are only needed during training the teacher network. Remarkably, with as few as 80 training meshes from standard datasets, our pipeline in the end achieves state-of-the-art results on matching raw point clouds which can be *non-isometric* (DT4D-H), *heterongeous* (SHREC07-H), or *with topological noise* (TOPKIDS).

---

### Author Rebuttal · Authors · 2023-08-09

We would like to thank all the reviewers for their time, effort, and insightful comments on our manuscript. We are glad that all the reviewers recognize our promising results on various benchmarks. We are encouraged by the recognition that our formulation is reasonable and that our writing is clear and easy to follow (Reviewer xFDL, CNuX, GUi9). We appreciate the recognition of our method's motivation and design by Reviewer CNuX, GUi9. We extend our gratitude to Reviewer GUi9 for acknowledging the difficulty of our task of interest and recognizing the contributions of our work to the shape matching community.

Before we clarify our main contributions and address some common concerns, we would like to kindly refer all the reviewers to our **Supp. Mat.**, in which we put a decent amount of extra experimental results and analysis, as well as a video clip demonstrating intuitively our pipeline. We also report several experimental results per reviewers' requests in the attached document, which we refer as the **Rebuttal Mat.** in the following.

### **Problem statement and key challenges:**
In this paper, we propose a hybrid pipeline for computing dense correspondences between a pair of unstructured point clouds, which can undergo significant non-rigid deformations. In particular, we encounter the following challenges:

1.	It is difficult to directly infer intrinsic structure (i.e., geometric structure regarding the underlying surface) from raw point clouds. A common practice is to perform meshing or to build certain graph Laplacian on top of them [8], which can be inefficient and unscalable;

2.	Non-rigid shape registration techniques [27, 45] can handle small to moderate non-rigid deformations via proximities in the ambient space R^3. However, in the presence of large deformation, they can fail significantly since the intermediate maps induced by Euclidean proximities do not necessarily respect the intrinsic, non-rigid deformations;

3.	Except for the purely intrinsic methods, most of the existing works either require the input point clouds to be extrinsically aligned [8, 22], or depend on heavy correspondence labels during training [30].

### **Novelty:**
To address the above challenges, we propose a systematic pipeline, which combines the learning-based shape matching and the optimization-based shape registration techniques. We conclude our key novelties as follows:

1.	To the best of our knowledge, we are the first to address the problem of **unstructured** point cloud matching via a hybrid approach. Relevant works such as Deep Shells, NeuroMorph strictly require meshes as input. We believe that lifting this assumption is novel and non-trivial. On the other hand, mesh-based methods can be sensitive to missing parts and topological noise (see Fig. 2, 3 & Table 1 in the Supp. Mat.);

2.	Our solution to the first challenge above is novel. We consider the most relevant work as SSMSM [8], which also follows a self-supervised approach. However, [8] essentially depends on the fact that DiffusionNet can be trained jointly over meshes and the inherent vertices. In fact, [8] explicitly constructs graph Laplacian during training and inference, which is less efficient. For instance, it takes more than one minute to pre-process an input of 40,000 points with a V100 GPU. In contrast, our self-supervised scheme is without pre-process and can in principle take any pair of backbones tailored for meshes and point clouds respectively. The natural and simple teacher-student learning scheme effectively learns a point-based feature extractor by mimicking the mesh-based counterpart. Unlike [8], which is adhered to DiffusionNet, our scheme is more general, flexible, and can be extended in the future (e.g., replacing DGCNN with a rotation-invariant point feature extractor).

### **Potential contributions to the community:**

We sincerely thank all the reviewers for their constructive comments, which help us to re-think and position our approach in a clearer way. In particular, we would like to highlight the following features:

1.	The idea of combining DFM prior and optimization-based shape registration techniques is simple and of great potential. Our approach is simply a first step along this line of exploration, which can undoubtedly benefit from the rapid advances from both directions;

2.	Our approach is **scalable** w.r.t input size. Thanks to the fact that we non-rigidly align shapes in R^3, we can in theory freely down- and up-sample both the template mesh and the target point clouds. Note this is non-trivial for methods based on mesh or graph Laplacian, as deducing dense maps with landmark correspondences over graph structures is a difficult task on its own [a]. In Fig. 4 of our Rebuttal Mat., we show the matching results on the real scans from the FAUST challenge [b], each of which consists of around 160,000 vertices. In contrast, [8] can handle at most 50,000 vertices without running out of 32G memory on a V100 GPU. We visualize its matching results on the subsampled (to 40,000 vertices) point clouds for comparison;

3.	As a natural follow-up, we in fact have managed to get some preliminary results on extending our pipeline to matching **partial** point clouds (see Fig. 1 in the Rebuttal Mat.). Essentially, we train a DFM tailored for partial-view point clouds generated on SCAPE_r dataset, and replace the two-way Chamfer distance with a one-way one. We also test our pipeline with noisy partial scans from [c] (see Fig. 2 in the Rebuttal Mat.). Though currently, it remains a challenge to deal with partial point clouds of arbitrary orientation and position, we believe the results sufficiently show the potential of our general scheme.

[a]: Weighted averages on surfaces, D. Panozzo, I. Baran, O. Diamanté, O. Sorkin-Hornung, SIGGRAPH 2013. [b]: https://faust-leaderboard.is.tuebingen.mpg.de [c]: http://domedb.perception.cs.cmu.edu

---

### Decision · Program_Chairs · 2023-09-21

**Decision:**

Accept (poster)

**Comment:**

The paper tackles unsupervised deformable shape matching and registration (alignment), an open and challenging problem across several communities. It received overall positive reviews initially, identifying key strengths in the extensive and convincing experimental results, and in the quality of the writing. Some potential weaknesses were reported on the potentially limited novelty of the approach, but these were resolved with an extensive rebuttal that was evaluated positively. We believe the paper to make further steps forward for the registration problem, and its quality meets the NeurIPS standards; we recommend acceptance, with the understanding that the suggested revisions should be incorporated in the camera ready version.